# ALLEVIATING LABEL SHIFT THROUGH SELF-TRAINED INTERMEDIATE DISTRIBUTION: THEORY AND ALGORITHMS

## ABSTRACT

As an obstacle towards real-world problems with the changing environment, label shift, which assumes the source and target label marginal distributions differ, loosens the homogeneous distribution assumption in classical learning scenarios. To correct the label shift, importance weighting is one of the most popular strategies with rigorous theoretical guarantees. However, the importance weight estimation of most existing methods results in high variance under large label shift or few source samples. In this paper, we introduce an intermediate distribution instead of the source distribution to reduce the variation to the target label distribution. Our approach learns a self-trained intermediate distribution constructed from the labeled source and unlabeled target samples to approximate the intermediate distribution. It balances the bias from pseudo target labels and the variance from importance weighting. Besides, we prove the sample complexity and generalization guarantees for our approach, which has a tighter generalization bound than the existing label shift methods under mild conditions. Extensive experimental results validate the effectiveness of our approach over existing state-of-the-arts methods.

## 1 INTRODUCTION

In the classical learning scenario, the model is usually trained on the labeled source set and predicted on the unlabeled target set, which is reasonable and effective when the source and target sets have independent and identically distributed samples Murphy (2012). However, the same distribution assumption is rarely satisfied in the real world Zhou (2022); Long et al. (2017); Wu et al. (2019); Quinonero-Candela et al. (2008). For example, in medical diagnostics, the disease distribution is often diverse in different regions, but the symptoms of the same disease are similar. In bird identification, the distribution of bird in spring is different from that in winter, but the appearance of birds does not change with the seasons. These phenomena are called label shift, where the proportion of categories differs between the source and target distribution ($p(Y) \neq q(Y)$), while the feature distribution is the same for each category ($p(X|Y) = q(X|Y)$) Hwang et al. (2022); Liu et al. (2022); Hong et al. (2022); Kim et al. (2022).

Owing to the change of label marginal distribution, the performance of traditional models is often reduced significantly Tasche (2017); Vaz et al. (2019); Tan et al. (2022); Ma et al. (2022). Therefore, in order to improve the model generalization ability, it is necessary to adjust traditional models with the distribution characteristic of label shift assumption. Most advanced label shift methods estimate the importance weights to reweight the source classifier and make it fit the target samples Podkopaev & Ramdas (2021); Azizzadenesheli (2022); Zhao et al. (2021); Zhang et al. (2021). For example, BBSE Lipton et al. (2018) is proposed to correct the source classifier through importance-weighted empirical risk minimization and guarantee the generalization error bound of any reweighted black box classifiers theoretically. On this basis, RLLS Azizzadenesheli et al. (2019) is proposed to obtain good statistical guarantees without a requirement on the problem-dependent minimum sample complexity as necessary for BBSE. Furthermore, RLLS designs a regularized weight estimator to reduce the influence of small target samples case on weight estimation and derives the corresponding generalization error bound with finite samples. The rest of label shift methods introduce maximum likelihood estimation or confidence calibration to correct for label shift that does not require model

retraining. MLLS Alexandari et al. (2020) combines maximum likelihood with bias-corrected temperature scaling, which outperforms both BBSE and RLLS across diverse datasets and distribution shifts. In addition, MLLS introduces a principled strategy for estimating source-domain priors that improves robustness to poor calibration. Unfortunately, although this kind of methods has strong reusability, it does not have an effective generalization guarantee in theory. Besides, the theoretical results show that the importance weight estimation and classifier generalization ability of BBSE and RLLS have high variance under large label shift or few source samples Lipton et al. (2018); Azizzadenesheli et al. (2019).

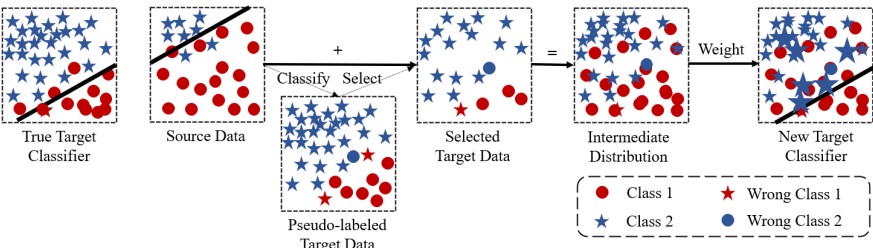

Figure 1: A flow presentation of ALS. Black lines depict the empirical risk minimizer.

In this paper, we introduce an intermediate distribution instead of the source distribution, which is a linear combination of source and target distributions, to alleviate label shift theoretically. As shown in Figure 1, our approach, abbreviated as ALS, learns a self-trained intermediate distribution constructed from the labeled source and unlabeled target samples to approximate the intermediate distribution. ALS can reduce label distribution differences, so as to obtain more accurate importance weight estimation. At the same time, ALS also introduces bias from its use of pseudo target labels, which reduces the generalization performance of the reweighted model. We theoretically analyze the impact of weight estimation variance and pseudo-label bias induced by self-trained intermediate distribution on ALS performance.

In order to verify the effectiveness of ALS in improving BBSE and RLLS performance, we show classifier prediction accuracy and weight estimation error for some kinds of shifts on the CIFAR-10 and MNIST datasets. The experiment results show that ALS framework can achieve an improvement in prediction accuracy of nearly 8% for large shifts and lower weight estimation error for most cases over BBSE and RLLS. In addition, we verify that the intermediate distribution has smaller label shift compared to the source distribution and analyze the influence of the proportion of sample selection on the model performance.

**Contribution.** (1) Introduction of an intermediate distribution instead of the source distribution to alleviate label shift. (2) A principled algorithm ALS, aimed at learning a self-trained intermediate distribution to approximate the ideal distribution. (3) The generalization guarantee of ALS under label distribution differences and pseudo target labels. (4) Extensive experiments are conducted to validate the effectiveness and support our theoretical findings.

## 2 PROBLEM SETTING

**Reweighted Framework.** We use random variables $X \in \mathcal{X}$, $Y \in \mathcal{Y}$ to model the features and labels respectively, where $\mathcal{X} = \mathbb{R}^d$ and $\mathcal{Y}$ is a discrete domain equivalent to $\{1, 2, ..., K\}$. $d$ is the feature dimension and $K$ is the total number of classes. We let $(X^P, Y^P) = \{x_i, y_i\}_{i=1}^n$ as a labelled set drawn i.i.d. from the source distribution $P$ and $X^Q = \{x_i\}_{i=n+1}^{n+m}$ as an unlabelled set drawn i.i.d. from the target distribution $Q$. We use $p$ and $q$ to denote the probability density or mass function associated with $P$ and $Q$. We quantify the shift using the exponent of the second and infinite order Renyi divergence as follows

$$d(q||p) := \sum_{i=1}^{K} q(i) \frac{q(i)}{p(i)}, \quad and \quad d_\infty(q||p) := \sup_i \frac{q(i)}{p(i)}. \tag{1}$$

Under the label shift assumption, i.e., $p(X|Y) = q(X|Y)$ and $p(Y) \neq q(Y)$, if we give a hypothesis class $\mathcal{H}$ and a loss function $l : \mathcal{X} \times \mathcal{Y} \to \mathbb{R}^1$, the goal of label shift setting is to find a hypothesis $h \in \mathcal{H}$ which minimizes the following reweighted loss Garg et al. (2020):

$$\mathcal{L}(h) = \mathbb{E}_{(X,Y) \sim Q}[l(h(X), Y)] = \mathbb{E}_{(X,Y) \sim P}[w(Y)l(h(X), Y)]. \tag{2}$$

Where $w \in \mathbb{R}^K$ is the importance weight and $w(Y) = q(Y)/p(Y)$. If $P = Q$, which represents that the source domain is same as the target domain, i.e., $w(Y) = \mathbf{1}$, the above problem degenerates to the standard learning paradigm.

In practice, the weight $w$ is unknown. Thus we minimize the following empirical loss with the estimated weight $\hat{w}$ vicariously:

$$\hat{\mathcal{L}}_P(h; \hat{w}) = \frac{1}{n} \sum_{i=1}^n \hat{w}(y_i)l(h(x_i), y_i). \tag{3}$$

**Importance Weight Estimation.** Assume $C_h$ is the confusion matrix with $[C_h]_{ij} = p(h(X^P) = i, Y^P = j)$ and $q_h$ is the vector which represents the probability mass function of $h(X^Q)$. In the usual finite sample setting, $\hat{C}_h$ and $\hat{q}_h$ corresponding to $C_h$ and $q_h$ are estimated by the existing samples. BBSE estimates the importance weight $\hat{w} = \hat{C}_h^{-1}\hat{q}_h$. RLLS calculates the weight shift degree $\hat{\theta} = \arg\min_\theta ||\hat{C}_h\theta - (\hat{q}_h - \hat{C}_h\mathbf{1})||_2 + \Delta_C \|\theta\|_2$, and obtains the important weight $\hat{w} = \gamma\hat{\theta} + \mathbf{1}$. To analyze the theoretical properties, we summarize the generalization bounds for the above two methods by the following unified form briefly.

**Corollary 1** *Azizzadenesheli et al. (2019) Assume that $\hat{h}_{\hat{w}}$ and $h^*$ are the solved and optimal classifier respectively , the following generalization bound holds with probability at least $1 - 2\delta$*

$$\mathcal{L}(\hat{h}_{\hat{w}}) - \mathcal{L}(h^*) \leq \varepsilon_{\mathcal{G}}\left(\frac{1}{n}, d_\infty(q||p), d(q||p)\right) + (1 - \gamma)\|w - 1\|_2 + \gamma\varepsilon_w\left(\frac{1}{n}, \|w - 1\|_2\right). \tag{4}$$

*where*

$$\varepsilon_{\mathcal{G}}\left(\frac{1}{n}, d_\infty(q||p), d(q||p)\right) := 2\mathcal{R}(\mathcal{G}(\mathcal{L}, \mathcal{H})) + \frac{2d_\infty(q||t)\log(2/\delta)}{n} + \sqrt{\frac{2d(q||t)\log(2/\delta)}{n}},$$

$$\varepsilon_w\left(\frac{1}{n}, \|w - 1\|_2\right) := \mathcal{O}\left(\frac{1}{\sigma_{\min}}\left(\|w - 1\|_2\sqrt{\frac{\log(K/\delta)}{n}} + \sqrt{\frac{\log(1/\delta)}{n}} + \sqrt{\frac{\log(1/\delta)}{m}}\right)\right),$$

*In addition, $\mathcal{G}(\mathcal{L}, \mathcal{H}) = \{g_h(x, y) = w(y)l(h(x), y), \ h \in \mathcal{H}\}$ and the Rademacher complexity is defined as $\mathcal{R}(\mathcal{G}(\mathcal{L}, \mathcal{H})) := \mathbb{E}_{(x_i, y_i) \in T}\left[\mathbb{E}_{\xi_i} \frac{1}{n}\left[\sup_{h \in \mathcal{H}} \sum_{i=1}^n \xi_i\hat{\pi}(x_i)g_h(x_i, y_i)\right]\right]$. $\sigma_{\min}$ is the minimum eigenvalue of confusion matrix $\hat{C}_h$, $\gamma \in (0, 1)$ is the balance parameter and $\mathcal{O}$ hides universal constant factors.*

**Intermediate Distribution.** By the above corollary, the larger label shift deviant $||w - 1||_2$, the lower model generalization performance. Thus, the concept of intermediate distribution is introduced to alleviate label shift.

**Definition 1** *(Intermediate Distribution). Give a distribution $T$ and $t$ is the corresponding probability density or mass function. If the label conditional distribution stays the same ($p(X|Y) = t(X|Y) = q(X|Y)$) and the label marginal distribution shift is alleviated ($|q(Y)/t(Y) - 1| < |q(Y)/p(Y) - 1|$), $T$ is taken to be the intermediate distribution.*

Interestingly enough, we find sufficient conditions for satisfying $T$ to be an intermediate distribution. Afterwards, given that some target labels are known, we construct a concrete set that follows the distribution $T$.

**Proposition 1** *If a distribution $T$ is a linear combination of source and target domain distributions, i.e., $T = \lambda P + (1 - \lambda)Q$ and $0 < \lambda < 1$, the distribution $T$ must be the intermediate distribution.*

**Proposition 2** *Assume that the $\hat{m}$ target labels are known, i.e., $(X^Q, Y^Q) = \{x_i, y_i\}_{i=n+1}^{n+\hat{m}} \sim Q$. Then we construct a new labeled set $(X^T, Y^T) = [(X^P, Y^P), (X^Q, Y^Q)]$, which follows the intermediate distribution $T$ with $\lambda = n/(n + \hat{m})$. If the classifier is trained on the new training set $(X^T, Y^T)$ using BBSE or RLLS, the generalization bound will be tighter.*

Due to the space limitation, we list the proof details in the supplementary material. In the above, we give a concrete example following the intermediate distribution to show its effect on the generalization bound theoretically. However, since the target samples are not labeled in practice, it is not straightforward to construct the intermediate distribution.

## 3 PROPOSED APPROACH

In this section, we make a self-trained distribution to approximate the intermediate distribution. Our approach can be divided into three parts. (1) Approximate partial true target labels $\{y_i\}_{i=n+1}^{n+\hat{m}}$ with pseudo labels $\{\hat{y}_i\}_{i=n+1}^{n+\hat{m}}$ with higher confidence to ensure the accuracy of pseudo labels. (2) Correct for the deviation of label conditional distribution which caused by the selected pseudo-label samples. (3) Use the new training set $(X^T, Y^T) = [\{x_i, y_i\}_{i=1}^n, \{x_i, \hat{y}_i\}_{i=n+1}^{n+\hat{m}}]$ which follows the self-trained distribution to retrain the classifier.

### 3.1 CONFIDENCE-GUIDED PSEUDO LABELS

For clarity of notation, we define the $k$-th element of label $y$ as $y^k$, the classifier parameters as $\upsilon$ and $\hat{p}(k|x, \upsilon)$ as the softmax probability for the $k$-th class. We employ the cross-entropy loss and estimate the importance weight $\hat{w}$ by BBSE or RLLS. Then by the reweighted framework, we can obtain the target classifier $\tilde{\upsilon}$ as follows:

$$\tilde{\upsilon} = \arg\min_{\upsilon} -\frac{1}{n} \sum_{i=1}^{n} \sum_{k=1}^{K} \hat{w}(k) y_i^k \log \hat{p}(k|x_i, \upsilon). \tag{5}$$

Experience shows that large soft probabilities lead to pseudo labels with high confidence. Thus, we propose the confidence-guided pseudo labels selection framework.

$$\begin{cases} \hat{Y}^Q = \arg\min_{y_i} -\frac{1}{m} \sum_{i=n+1}^{n+m} \sum_{k=1}^{K} y_i^k \log \frac{\hat{p}(k|x_i, \tilde{\upsilon})}{\lambda_k}, \\ s.t. \quad y_i^k \in \{0, 1\}, \ \forall i \in \{n+1, ..., n+m\}. \end{cases} \tag{6}$$

The global optimal solution of the above problem is given as following:

$$\hat{y}_i^{k^*} = \begin{cases} 1, \text{ if } k^* = \arg\max_{k} \left\{ \frac{\hat{p}(k|x_i, \tilde{\upsilon})}{\lambda_k} \right\} \text{ and } \hat{p}(k|x_i, \tilde{\upsilon}) > \lambda_k, \\ 0, \text{ otherwise.} \end{cases} \tag{7}$$

Note that $\hat{y}_i^*$ can be not only a one-hot, but also an all-zero vector, depending on whether the corresponding sample $\hat{x}_i$ is selected. If $\hat{y}_i^*$ is a one-hot, the sample is selected and if $\hat{y}_i^*$ is an all-zeros vector, the sample is not chosen. Specifically, we can regard the probability of the $k$-th class $\hat{p}(k|x_i, \tilde{\upsilon})$ as the confidence of the prediction. If a predication satisfies $\hat{p}(k|x_i, \tilde{\upsilon}) \geq \lambda_k$, the sample is selected with confidence and classified as $k^* = \arg\max_{k} \left\{ \frac{\hat{p}(k|x_i, \tilde{\upsilon})}{\lambda_k} \right\}$. On the other hand, the less confident samples corresponding to $\hat{p}(k|x_i, \tilde{\upsilon}) \leq \lambda_k$ are not selected.

$\{\lambda_k\}_{k=1,..,K}$ are critical parameters to control pseudo-label learning and selection. Drawing on the traditional approaches introduced in Zou et al. (2018) and Zou et al. (2019), we introduce a strategy of determining $\{\lambda_k\}_{k=1,..,K}$ in our scenario. Specifically, we attempt to select samples which satisfy the marginal probability $q(Y)$ with high confidence. Since the true $q(Y)$ is unknown and we calculate it by $\hat{q}(Y) = \hat{w}p(Y)$. We only use one ratio $r$ to determine how many target samples to select for all classes. For the $k$-th class, the number of selected samples is $\hat{m}_k = [rm\hat{q}(Y_k)]$, where $[.]$ is the rounding function. Thus $\{\lambda_k\}$ is set as the prediction with the lowest confidence in the $\hat{m}_k$-th predictions of class $k$.

## 3.2 Alignment of label conditional distributions

Suppose that the selected pseudo-labeled data follows the distribution $\hat{Q}$. Due to selection bias, the label conditional distribution may change, i.e.$\hat{q}(X|Y) \neq q(X|Y)$, which violates label shift assumption. This results in an impact on the expected loss of the intermediate distribution as Proposition 3.

**Proposition 3** *Assume that the classifier has an upper bound loss, that is, there exists at least one $h$ satisfying $l(h; (X, Y) \in \{P, T, Q\}) \leq \alpha^*$. Let $\mathbb{E}_Q[l(h)]$ represents $\mathbb{E}_{X,Y \in Q}[l(h(X), Y)]$ for convenience. Under the $\alpha^*$-loss assumption, for learning model $\{l, h\}$, the following results hold.*

$$
\begin{cases}
|\mathbb{E}_P[l(h)] - \mathbb{E}_Q[l(h)]| \leq 2\alpha^* d_{TV}(p(Y), q(Y)), \\
|\mathbb{E}_T[l(h)] - \mathbb{E}_Q[l(h)]| \leq 2\alpha^* d_{TV}(t(Y), q(Y)) + 2\alpha^* \min\{\mathbb{E}_Q(D_{tq}(X, Y)), \mathbb{E}_T(D_{tq}(X, Y))\},
\end{cases}
$$
(8)

*where $d_{TV}(.,.)$ represents the Total Variation (TV) distance between probability distributions and $D_{tq}(X, Y) = d_{TV}(t(X|Y), q(X|Y))$.*

Through the above theorem, it is obvious that even if $d_{TV}(t(Y), q(Y)) < d_{TV}(p(Y), q(Y))$, it is hard to guarantee that the upper bound of expected difference satisfies $\sup |\mathbb{E}_T[l(h)] - \mathbb{E}_Q[l(h)]| < \sup |\mathbb{E}_P[l(h)] - \mathbb{E}_Q[l(h)]|$ due to the TV distance $D_{tq}(X, Y)$. However, since $D_{tq}(X, Y)$ is difficult to optimize directly, we give a sufficient condition in Proposition 4.

**Proposition 4** *If the conditional distribution satisfies $\hat{q}(X|Y) = p(X|Y)$, $D_{tq}(X, Y) = 0$ holds.*

We employ Kernel Mean Matching Yu & Szepesvári (2012) to align conditional distributions and minimize the Maximum Mean Discrepancy Cui et al. (2020) loss: $\sum_{k=1}^{K} \left\| E_{X \sim p(X|Y_k)}[\varphi(X)] - E_{X \sim \hat{q}(X|Y_k)}[\varphi(X)\pi^k] \right\|^2$, where $\pi^k \in \mathbb{R}^{\hat{m}_k}$ is the in-class weight and $\varphi(\cdot)$ is a kernel mapping. We approximate the expected kernel mean values by the empirical ones:

$$
\min_{\pi} \sum_{k=1}^{K} \left\| \frac{1}{\hat{m}_k} \sum_{i=1}^{\hat{m}_k} \pi_i^k \varphi(x_i^k) - \frac{1}{n_k} \sum_{j=1}^{n_k} \varphi(x_j^k) \right\|^2.
$$
(9)

Suppose that the true weight is $\pi_* = [\pi_*^1; ...; \pi_*^K] \in \mathbb{R}^{\hat{m}}$. The above loss can be reparametrized as

$$
\min_{\pi} \sum_{k=1}^{K} \left( \frac{(\pi^k)^T \mathbf{K}^{\hat{Q}} \pi^k}{\hat{m}_k^2} - 2 \frac{1^T \mathbf{K}^{P,\hat{Q}} \pi^k}{\hat{m}_k n_k} \right),
$$
(10)

where $\mathbf{K}^{\hat{Q}}$ is the kernel matrix of $X^{\hat{Q}}$ and $\mathbf{K}^{P,\hat{Q}}$ is the cross kernel matrix. In this paper, the Gaussian kernel, i.e., $\mathbf{K}(i, j) = \exp\left(-\frac{||x_i - x_j||^2}{2\sigma^2}\right)$ is applied, where $\sigma$ is the bandwidth.

In order to ensure that as many samples as possible are selected from the target set, we add a regularization term for $\{\pi^k\}_{k=1}^K$. However, sample weights $\pi^k$ may change the marginal distribution, which violates the condition $|w_t - 1| \leq |w_q - 1|$. Thus we introduce a constraint for sample weight $\{\pi^k\}_{k=1}^K$, which theoretically guarantees the invariance of marginal distribution, as shown in Proposition 5.

**Proposition 5** *Assume that $\pi_i^k$ is the i-th of $\pi^k$, which corresponds to the weight of the i-th data of the k-th class. If $\sum_{i=1}^{\hat{m}_k} \pi_i^k = \hat{m}_k$ holds for each class $k$, then $\hat{q}_\pi(Y) = \hat{q}(Y)$ is established, where $\hat{q}_\pi(.)$ is the weighted pseudo-label distribution.*

Combined with the above information, the final loss changes to the following form:

$$
\begin{cases}
\min_{\pi} \sum_{k=1}^{K} \left( \frac{(\pi^k)^T \mathbf{K}^{\hat{Q}} \pi^k}{\hat{m}_k^2} - 2 \frac{1^T \mathbf{K}^{P,\hat{Q}} \pi^k}{\hat{m}_k n_k} + \beta_k \|\pi^k\|^2 \right), \\
s.t. \sum_{i=1}^{\hat{m}_k} \pi_i^k = \hat{m}_k, \ \pi_i^k \geq 0, \ \forall k \in \{1, ..., K\}.
\end{cases}
$$
(11)

The optimal solution of the above problem is $\hat{\pi} \in \mathbb{R}^{\hat{m}}$, which can be obtained by Quadratic Programming(QP). Let $\mathcal{D}(\pi) = \sum_{k=1}^{K} \left\| E_{X \sim p(X|Y_k)}[\varphi(X)] - E_{X \sim \hat{q}(X|Y_k)}[\varphi(X)\pi^k] \right\|^2$, we analyze the upper bound of $\mathcal{D}(\hat{\pi}) - \mathcal{D}(\pi_*)$, which represents the difference between distributions $\hat{q}(X|Y)$ and $p(X|Y)$.

**Proposition 6** *Let the kernel mapping be universal and upper bounded that $\|\varphi(x)\| \leq \mu$ for the intermediate set. Through Proposition 5, it is obvious that $\|\pi^k\|_1 = \hat{m}_k$. With probability at least $1 - \delta$, we have*

$$\mathcal{D}(\hat{\pi}) - \mathcal{D}(\pi_*) \leq \varepsilon_\pi(n, \hat{m}) = 16\mu^2 \sum_{k=1}^{K} \sqrt{2\left(\frac{1}{\hat{m}_k} + \frac{1}{n_k}\right) + 2\sqrt{2\left(\frac{1}{n_k} + \frac{1}{\hat{m}_k}\right) \log \frac{K}{\delta}}}. \quad (12)$$

### 3.3 THE RETRAINING OF TARGET CLASSIFIER

Once sample weights $\hat{\pi}(x)$ are obtained, we re-estimate the importance weights $\hat{w}_t$ by BBSE or RLLS on the new training set $[\{x_i, y_i\}_{i=1}^n, \hat{\pi}(x_i)\{x_i, \hat{y}_i\}_{i=n}^{n+\hat{m}}]$. Thus, the form of the final target empirical loss is given below:

$$\widehat{\mathcal{L}}(h; \widehat{w}_t, \hat{\pi}) = \frac{1}{n + \hat{m}} \left( \sum_{i=1}^n \hat{w}_t(y_i) l(h(x_i), y_i) + \sum_{i=n}^{n+\hat{m}} \hat{w}_t(y_i) \hat{\pi}(x_i) l(h(x_i), \hat{y}_i) \right). \quad (13)$$

Assume that the deviation form of label conditional and marginal distribution is $d_{X|Y}(q||\hat{q}) := \mathbb{E}_{X,Y \in Q}\left[\left|\left|\frac{q(X|Y) - \hat{q}(X|Y)}{t(X|Y)}\right|\right|\right]$ and $d_Y(q||\hat{q}) := \mathbb{E}_{X,Y \in Q}\left[\left|\left|\frac{q(Y) - \hat{q}(Y)}{\hat{t}(Y)}\right|\right|\right]$. Traditional generalization theory 1 can not apply to our problem directly due to conditional distribution deviation and pseudo label noise. Thus we propose a new generalization error bound to balance the pseudo label bias and importance weighting variance.

**Proposition 7** *We assume that the true importance weight is $\tau(X, Y) := \frac{q(X,Y)}{t(X,Y)}$ and the pseudo importance weight is $w_t(Y)$. In the case of ALS framework based on RLLS, with probability at least $1 - 3\delta$, the ALS generalizes as:*

$$\mathcal{L}(\hat{h}_{\hat{w}_t}; \tau) - \mathcal{L}(h^*; \tau) \leq \varepsilon_{\mathcal{G}}(n + \hat{m}, d_\infty(q||t), d(q||t)) + \varepsilon_w(n + \hat{m}, ||w - 1||_2)$$
$$+ \frac{2\alpha^* rm}{n + rm}\left(d_{X|Y}(q||\hat{q}) + d_Y(q||\hat{q})\right) + 2\alpha^* \|w_t(Y)\|_\infty err(h_{\tilde{v}}). \quad (14)$$

*Where $\varepsilon_{\mathcal{G}}(n + \hat{m}, d_\infty(q||t), d(q||t)) = 2\mathcal{R}(\mathcal{G}(\mathcal{L}, \mathcal{H})) + \frac{2\alpha^* d_\infty(q||t) \log(2/\delta)}{n+\hat{m}} + \sqrt{\frac{2\alpha^* d(q||t) \log(2/\delta)}{n+\hat{m}}}$,*
*$\varepsilon_w(n + \hat{m}, ||w - 1||_2) = 2\lambda\alpha^* \mathcal{O}\left(\frac{1}{\sigma_{\min}}\left(||w - 1||_2 \sqrt{\frac{\log(K/\delta)}{n+\hat{m}}} + \sqrt{\frac{\log(1/\delta)}{n+\hat{m}}} + \sqrt{\frac{\log(1/\delta)}{m}}\right)\right) + 2\alpha^*(1 - \lambda)||w - 1||_2$, $err(h_{\tilde{v}})$ denotes the importance weighted 0/1-error of the predictor $h_{\tilde{v}}$.*

## 4 EXPERIMENT

### 4.1 EXPERIMENTAL SETUP

**Datasets.** We evaluate the performance and effectiveness of ALS on the numerous artificial shifts on the MNIST Li (2012), CIFAR10 Krizhevsky et al. (2009) and CIFAR100 datasets. In our experiment, every dataset is randomly divided into two parts which represent the source and target domain of the same size. Then we sample data points as the source and target set from each domain. For MNIST, we set the number of source and target set as 2000 and 8000 respectively. Similarly, we set the size of source and target set as 3000 and 10000 for CIFAR10 and the size of source and target set as 20000 and 30000 for CIFAR100. Two types of shifts are considered in our experiments: the Tweak-One shift refers to the case that a class changes to the specified probability $\rho$ which is set to $\{0.2, 0.5, 0.8\}$(larger values of $\rho$ result in more extreme label shift), while the distribution of other classes remained constant. The Dirichlet shift generates the class proportions by the dirichlet distribution with the concentration parameter $\alpha$ which is set to $\{1, 2, 5\}$(smaller values of $\alpha$ result in

Table 1: Acc comparison with different ratio $r$ on Dirichlet shift datasets.

| Methods | $\alpha = 1$ | | | $\alpha = 2$ | | | $\alpha = 5$ | | |
|---|---|---|---|---|---|---|---|---|---|
| | $r=0.1$ | $r=0.3$ | $r=0.5$ | $r=0.1$ | $r=0.3$ | $r=0.5$ | $r=0.1$ | $r=0.3$ | $r=0.5$ |
| MNIST | | | | | | | | | |
| WW | 0.6908 | 0.6908 | 0.6908 | 0.7663 | 0.7663 | 0.7663 | 0.8075 | 0.8075 | 0.8075 |
| CBST | 0.7688 | 0.7865 | 0.7843 | 0.8186 | 0.8244 | 0.8259 | 0.8547 | 0.8561 | 0.8583 |
| BBSE | 0.7635 | 0.7635 | 0.7635 | 0.8018 | 0.8018 | 0.8018 | 0.8386 | 0.8386 | 0.8386 |
| RLLS | 0.7734 | 0.7734 | 0.7734 | 0.8058 | 0.8058 | 0.8058 | 0.8416 | 0.8416 | 0.8416 |
| BBSE-ALS | 0.8339 | 0.8394 | 0.8262 | 0.8535 | 0.8592 | 0.854 | 0.8662 | 0.8708 | 0.8736 |
| RLLS-ALS | 0.8405 | 0.8424 | 0.8337 | 0.8547 | 0.8599 | 0.8542 | 0.8645 | 0.8700 | 0.8746 |
| AvgIMP | **0.0687** | **0.07245** | **0.0615** | **0.0503** | **0.0588** | **0.0503** | **0.0253** | **0.0303** | **0.0340** |
| CIFAR10 | | | | | | | | | |
| WW | 0.2829 | 0.2829 | 0.2829 | 0.3519 | 0.3519 | 0.3519 | 0.3705 | 0.3705 | 0.3705 |
| CBST | 0.3380 | 0.3536 | 0.3502 | 0.3946 | 0.4008 | 0.3971 | 0.4258 | 0.4392 | 0.4336 |
| BBSE | 0.3476 | 0.3476 | 0.3476 | 0.3968 | 0.3968 | 0.3968 | 0.4036 | 0.4036 | 0.4036 |
| RLLS | 0.3609 | 0.3609 | 0.3609 | 0.4007 | 0.4007 | 0.4007 | 0.4104 | 0.4104 | 0.4104 |
| BBSE-ALS | 0.4220 | 0.4354 | 0.4299 | 0.4362 | 0.4396 | 0.4465 | 0.4411 | 0.4565 | 0.4621 |
| RLLS-ALS | 0.4354 | 0.4398 | 0.4328 | 0.4303 | 0.4412 | 0.4482 | 0.4426 | 0.4627 | 0.4615 |
| AvgIMP | **0.0743** | **0.0834** | **0.0771** | **0.0345** | **0.0417** | **0.0486** | **0.0349** | **0.0526** | **0.0548** |
| CIFAR100 | | | | | | | | | |
| WW | 0.2745 | 0.2745 | 0.2745 | 0.3817 | 0.3817 | 0.3817 | 0.4058 | 0.4058 | 0.4058 |
| CBST | 0.3348 | 0.3472 | 0.3488 | 0.4421 | 0.4537 | 0.4553 | 0.4781 | 0.4892 | 0.4854 |
| BBSE | 0.3365 | 0.3365 | 0.3365 | 0.4513 | 0.4513 | 0.4513 | 0.4877 | 0.4877 | 0.4877 |
| RLLS | 0.3482 | 0.3482 | 0.3482 | 0.4556 | 0.4556 | 0.4556 | 0.4932 | 0.4932 | 0.4932 |
| BBSE-ALS | 0.4016 | 0.4054 | 0.4098 | 0.4734 | 0.4771 | 0.4827 | 0.5004 | 0.5148 | 0.5163 |
| RLLS-ALS | 0.4075 | 0.4142 | 0.4156 | 0.4901 | 0.4935 | 0.4964 | 0.5122 | 0.5214 | 0.5236 |
| AvgIMP | **0.0622** | **0.0675** | **0.0704** | **0.0283** | **0.0319** | **0.0361** | **0.0159** | **0.0277** | **0.0295** |

more extreme label shift). In addition, we discuss the effect of the number of pseudo-label samples on the model performance, that is, the ratio $r \in \{0.1, 0.3, 0.5\}$ on the target sets.

**Compared methods.** In the main experiment section, we show the performance of ALS by comparing several variations. (1) WW shows the performance of the base classifier if the estimated importance weights are not used. (2) CBST Zou et al. (2018) is an advanced self-training learning method. (3) BBSE and RLLS are two standard baselines. (4) BBSE-ALS and RLLS-ALS are two variations to demonstrate the performance of ALS framework when the base methods are BBSE and RLLS respectively. (5) AvgImp represents the average performance improvement of the ALS framework over RLLS and BBSE.

**Network architecture.** BBSE and RLLS can use any classifier as the base classifier. It is worth noting that the more accurate the classifier, the more accurate the estimated weight. In our main experimental setup, we use a two-layer fully connected neural network for MNIST and CIFAR10 datasets simultaneously, and use Resnet18 He et al. (2016) for CIFAR100 dataset.

**Evaluation indicators and parameter setting.** For each shift type, we randomly sample 10 times using the corresponding distribution parameters to evaluate the predictive Accuracy(Acc) and F-score on the target set, and the Mean Square Error of the estimated weights. For our ALS, we set the regularized parameter $\beta$ from $\{0, 0.001, 0.1, 10, 1000\}$ and the parameter $\lambda$ is the same as that in RLLS.

## 4.2 PREDICTIVE RESULTS

Firstly, to show the effectiveness of ALS framework, we report the classification Accuracy(Acc) under the different proportions of pseudo label samples on the Dirichlet shift datasets. Other results

are shown in the supplementary material. All methods run on framework with Pyhton 3.7 and Pytorch. As seen from the results in Table 1, we have the following observations.

(1) Although different compared methods have different performances on different data sets, our ALS framework outperforms other methods in all cases. For example, on the Dirichlet shift CIFAR10 with $\alpha = 1$, the Acc improvement is more than 8%. These results fully validate the validity of ALS framework. (2) It is worth noting that our approach achieves greater improvement in the case of large label shift. For example, on the CIFAR10 with $\alpha = 1$, the Acc improvement is about 8% and on the CIFAR100 with $\alpha = 5$, the Acc improvement is only 2%. This suggests that small label shift may have little effect on model performance, thus the importance weight of ALS correction is not effective significantly. This can also be illustrated by the comparison with CBST. (3) We find that under the large label shift, the experimental results do not always improve with the increase of the proportion of pseudo label samples selected. The reason is that with the increase of pseudo label samples, its true accuracy will decrease. This not only adds more error samples to the training process, but also affects the estimate of importance weight. (4) Compared with WW, all the label shift methods improve the predictive performance, especially in the case of large label shift. This fully demonstrates the effectiveness of the importance weight estimation strategy. (5) Sometimes Acc metric does not fully capture the accuracy of each category, especially small categories. Therefore, F-score metric is adopted in our paper, and the experimental results are shown in Figure 2.(d). The experimental results show that our approach is also effective in small categories.

### 4.3 PERFORMANCE ANALYSIS

**Weight estimation analysis.** Furthermore, to illustrate the effectiveness of our shift strategy, we show the Mean Square Error of the estimated weights, i.e. $||w - w^*||^2$. As seen from the results in Figure 2.(a,b,c), the performance of ALS is still better than other methods in most cases, especially in the small pseudo label selection ratio. These results fully validate the validity of ALS framework. But in Figure 2.(a), we find that if the pseudo label selection ratio is 50% with the large label shift, the weight error of ALS estimation is larger than BBSE and RLLS. This is because the large shift classifier leads to the increase of weight estimation error, coupled with the wrong label, resulting in poor ALS framework effect.

**Visualization of Label Distribution Bias.** In order to show that ALS can alleviate label shift through self-trained intermediate distribution, we illustrate some visualization result in Figure 2.(e,f), which shows the label shift degree $w - 1$ of the intermediate and source distributions. The experimental results show that the ALS framework can effectively reduce the label distribution difference, both at positive and negative shifts.

**The Effect of Pesudo Label Ratios.** In order to explore the influence of pseudo-label samples on model performance, we compare the predictive accuracy of the ALS framework, pseudo and true labels. We set Pseudo w as the weight under pseudo labels and True w as the weight under true labels. In addition, if $r = 0$, BBSE-ALS and RLLS-ALS degenerates into BBSE and RLLS. In Figure 2.(g,h), we find that ALS framework can achieve the performance of Pseudo w under different conditions roughly. This basically hits the performance ceiling of ALS framework, since the target labels are unknown.

## 5 RELATED WORK

**Label shift** is an important type of domain adaption Yu et al. (2020); Ben-David et al. (2006); des Combes et al. (2020); Zhang et al. (2013) and has attracted wide attention in recent years. Label shift arises when the source and target domain have different class distribution, but the same feature distribution of each class Le et al. (2021); Zhao et al. (2021); Podkopaev & Ramdas (2021); Wu et al. (2021). Existing solutions are divided into the following two steps basically: (1) estimate importance weight, (2) construct an unbiased estimate of the target risk. In Lipton et al. (2018) and Azizzadenesheli et al. (2019), the class-weighted samples are used to retrain a new model under the Empirical Risk Minimization(ERM) framework while the weight is estimated by the confusion matrix and the predicted target labels. Yu et al. propose a method that can reduce the side-effect of noisy source labels in the label shift scenario in Yu et al. (2020). However, existing label shift methods only use the unlabeled target samples to estimate the importance weight and do not use

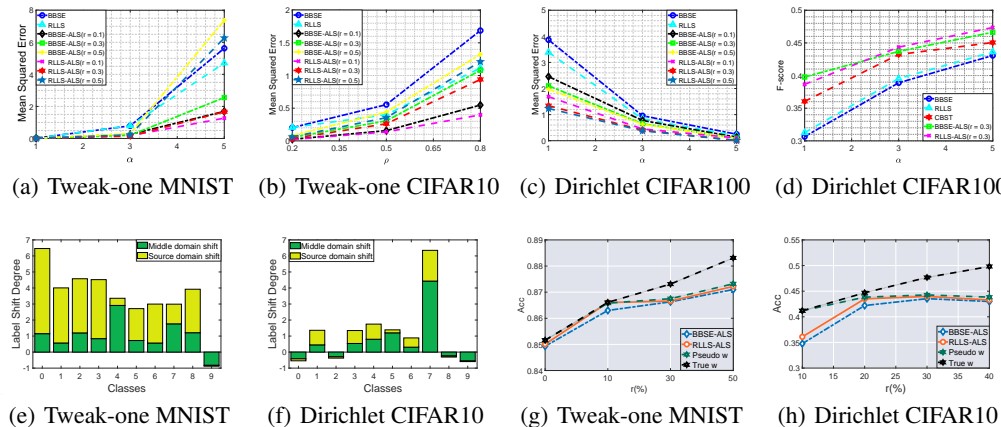

Figure 2: Various performance analysis of ALS.

them to train the target classifier directly, which is insufficient to use the sample information. In addition, our ALS framework can also be extended to other label shift methods without theoretical guarantee, which shows the extensibility of ALS framework.

**Pseudo label learning** depends on self-training Ke et al. (2022); Oymak & Gulcu (2021); Van Engelen & Hoos (2020); Zhou (2021), which trains the model on labeled samples and generates pseudo labels on unlabeled samples, and then uses the true and pseudo labels to retrain the model together Zoph et al. (2020); Xie et al. (2020); RoyChowdhury et al. (2019). The core of self-training is how to make the pseudo labels as accurate as possibleTai et al. (2021); Kumar et al. (2020); Wei et al. (2021). Zou et al.Zou et al. (2019) propose a confidence regularized self-training approach which can reduce the misleading effect caused by incorrect or fuzzy pseudo labels by using confidence-guide regularization to smooth output vectors. In Zhang et al. (2022), the generalization ability of self-training is evaluated when the classifier is a one hidden layer neural network. In addition, self-training is also widely used in domain adaptation. For example, the limitation of standard self-training under the condition of distribution deviation is overcome and its ability is verified in learning pseudo labels with cross-domain generalization in Liu et al. (2021). However, due to the difference of distribution shift hypothesis, the existing self-training methods for domain adaptation cannot solve our problem, which shows the creativity of ALS framework.

## 6 CONCLUSIONS

In this paper, we focus on how to make full use of target domain samples and propose a novel ALS framework to improve the performance of existing label shifting methods BBSE and RLLS. ALS framework leads to a more robust importance weight estimator as well as generalization guarantees under certain conditions. The experimental results show the superiority of ALS framework in both small and large label shifts. We believe that this work is an important step to improve the effect of label shift algorithms in practical applications because of its excellent performance. In future work, we plan to expand our approach to generalized label shift setting which imposes stricter requirements on conditional distribution alignment. However, how to keep the generalization theory in generalized label shift is an open problem and we will conduct in-depth exploration in future work.

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

# A APPENDIX

## A.1 PROOF OF PROPOSITION

### A.1.1 PROOF OF PROPOSITION 1

**Proposition 1** *If a distribution $T$ is a linear combination of source and target domain distributions, i.e., $T = \lambda P + (1 - \lambda)Q$ and $0 < \lambda < 1$, the distribution $T$ must be the intermediate distribution.*

**Proof:** If $T = \lambda P + (1 - \lambda)Q$, then $t(Y) = \lambda p(Y) + (1 - \lambda)q(Y)$ and $t(X|Y) = \lambda p(X|Y) + (1 - \lambda)q(X|Y)$. Under the assumption of label shift, i.e., $p(X|Y) = q(X|Y)$, then the label conditional distribution stays the same $(t(X|Y) = q(X|Y))$.

Due to $\left|\frac{q(Y)}{t(Y)} - 1\right| = \left|\frac{q(Y)}{\lambda p(Y) + (1-\lambda)q(Y)} - 1\right| = \left|\frac{\lambda(q(Y)-p(Y))}{\lambda p(Y) + (1-\lambda)q(Y)}\right| \leq \left|\frac{\lambda(q(Y)-p(Y))}{\lambda p(Y)}\right| = \left|\frac{q(Y)}{p(Y)} - 1\right|$, thus the label marginal distribution shift alleviates $(|q(Y)/t(Y) - 1| < |q(Y)/p(Y) - 1|)$. To summarize the above two points, $T$ is an intermediate distribution.

### A.1.2 PROOF OF PROPOSITION 2

**Proposition 2** *Assume that the $m$ target labels are known, i.e., $(X^Q, Y^Q) = \{x_i, y_i\}_{i=n+1}^{n+m} \sim Q$. Then we construct a new labeled set $(X^T, Y^T) = [(X^P, Y^P), (X^Q, Y^Q)]$, which follows the intermediate distribution $T$ with $\lambda = n/(n + \hat{m})$. If the classifier is trained on the new training set $(X^T, Y^T)$ using BBSE or RLLS, the generalization bound will be tighter.*

**Proof:** If $(X^T, Y^T) = [(X^P, Y^P), (X^Q, Y^Q)]$, Aasume that the number of samples of class $k$ in the source set is $n_k$ and the number of samples of class $k$ in the target set is $\hat{m}_k$. Then

$$
\begin{aligned}
t(X|Y_k) &= \mathbb{E}\left(\frac{1}{n_k + \hat{m}_k}\left(\sum_{i=1}^{n_k} x_i^P + \sum_{i=1}^{\hat{m}_k} x_i^Q\right)\right) \\
&= \frac{n_k}{n_k + \hat{m}_k}\mathbb{E}\left(\frac{1}{n_k}\sum_{i=1}^{n_k} x_i^P\right) + \frac{\hat{m}_k}{n_k + \hat{m}_k}\mathbb{E}\left(\frac{1}{\hat{m}_k}\sum_{i=1}^{\hat{m}_k} x_i^Q\right) \\
&= \frac{n}{n + \hat{m}}p(X|Y_k) + \frac{\hat{m}}{n + \hat{m}}q(X|Y_k), \qquad \forall k \in \{1, ..., K\}.
\end{aligned}
\tag{1}
$$

$$
t(Y_k) = \frac{n_k + \hat{m}_k}{n + \hat{m}} = \frac{n}{n + \hat{m}}\frac{n_k}{n} + \frac{\hat{m}}{n + \hat{m}}\frac{\hat{m}_k}{\hat{m}} = \frac{n}{n + \hat{m}}p(Y_k) + \frac{\hat{m}}{n + \hat{m}}q(Y_k), \forall k \in \{1, ..., K\}.
\tag{2}
$$

Thus the new training set $(X^T, Y^T)$ follows the intermediate distribution $T = \lambda P + (1 - \lambda)Q$ with $\lambda = n/(n + \hat{m})$. Next, we analyze why the generalization bound will be tighter.

[1]. It is obvious that the number of training samples changes from $n$ to $n + \hat{m}$, thus the number of training samples increases.

[2]. Assume $p(Y_k) \le q(Y_k)$, i.e., $\frac{n_k}{n} \le \frac{\hat{m}_k}{\hat{m}}$, then

$$
\begin{aligned}
p(Y_k) &= \frac{n_k}{n} = \frac{n_k + \frac{n_k}{n}\hat{m}}{n + \hat{m}} \\
&\le \frac{n_k + \frac{\hat{m}_k}{\hat{m}}\hat{m}}{n + \hat{m}} = \frac{n_k + \hat{m}_k}{n + \hat{m}} = t(Y_k) = \frac{\frac{n_k}{n}n + \hat{m}_k}{n + \hat{m}} \\
&\le \frac{\frac{\hat{m}_k}{\hat{m}}n + \hat{m}_k}{n + \hat{m}} = \frac{\hat{m}_k}{\hat{m}} = q(Y_k), \qquad \forall k \in \{1, 2, ..., K\}.
\end{aligned}
\tag{3}
$$

Thus, if $p(Y_k) \le q(Y_k)$, $p(Y_k) \le t(Y_k) \le q(Y_k)$ and vice versa. This leads to the following conclusion:

$$
\|w_p - 1\|_2 = \sqrt{\sum_{k=1}^{K}\left(\frac{q(Y_k)}{p(Y_k)} - 1\right)^2} \ge \sqrt{\sum_{k=1}^{K}\left(\frac{q(Y_k)}{t(Y_k)} - 1\right)^2} = \|w_t - 1\|_2.
\tag{4}
$$

Hence the label distribution deviation $\|w - 1\|_2$ shrunks.

[3]. If $p(Y_k) \le q(Y_k)$, then $\frac{q(Y_k)}{p(Y_k)} \ge \frac{q(Y_k)}{t(Y_k)} \ge 1$ holds and if $p(Y_k) \ge q(Y_k)$, then $\frac{q(Y_k)}{p(Y_k)} \le \frac{q(Y_k)}{t(Y_k)} \le 1$ holds, this leads to the following conclusion:

$$
\begin{cases}
d_\infty(q||p) = \sup_k \frac{q(Y_k)}{p(Y_k)} \ge \sup_k \frac{q(Y_k)}{t(Y_k)} = d_\infty(q||t) \ge 1, \\
d(q||p) = \sum_{k=1}^{K} q(Y_k)\frac{q(Y_k)}{p(Y_k)} \ge \sum_{k=1}^{K} q(Y_k)\frac{q(Y_k)}{t(Y_k)} = d(q||t).
\end{cases}
\tag{5}
$$

Thus, the infinite and second order Renyi divergence $d_\infty(q||p)$ and $d(q||p)$ reduce. Combining the above three conditions, the generalization bound will be tighter.

### A.1.3 PROOF OF PROPOSITION 3

**Proposition 3** *Assume that the classifier has an upper bound loss, that is, there exists at least one $h$ satisfying $l(h; (X, Y) \in \{P, T, Q\}) \le \alpha^*$. Let $\mathbb{E}_Q[l(h)]$ represents $\mathbb{E}_{X,Y \in Q}[l(h(X), Y)]$ for convenience. Under the $\alpha^*$-loss assumption, for learning model $\{l, h\}$, the following results hold.*

$$
\begin{cases}
|\mathbb{E}_P[l(h)] - \mathbb{E}_Q[l(h)]| \le 2\alpha^* d_{TV}(p(Y), q(Y)), \\
|\mathbb{E}_T[l(h)] - \mathbb{E}_Q[l(h)]| \le 2\alpha^* d_{TV}(t(Y), q(Y)) + 2\alpha^* \min\{\mathbb{E}_Q(D_{tq}(X, Y)), \mathbb{E}_T(D_{tq}(X, Y))\}.
\end{cases}
\tag{6}
$$

*where $d_{TV}(.,.)$ represents the Total Variation (TV) distance between probability distributions and $D_{tq}(X, Y) = d_{TV}(t(X|Y), q(X|Y))$.*

**Proof:** Due to $p(X|Y) = q(X|Y)$, we have

$$|\mathbb{E}_P[l(h)] - \mathbb{E}_Q[l(h)]|$$

$$= \left| \int_X \int_Y l(h)p(X = x, Y = y)dxdy - \int_X \int_Y l(h)q(X = x, Y = y)dxdy \right|$$

$$\le \alpha^* \int_X \int_Y |p(X = x|Y = y)p(Y = y) - q(X = x, Y = y)q(Y = y)| \, dxdy$$

$$= \alpha^* \int_X \int_Y p(X = x|Y = y) |p(Y = y) - q(Y = y)| \, dxdy \qquad (7)$$

$$= \alpha^* \int_Y |p(Y = y) - q(Y = y)| \int_X p(X = x|Y = y)dxdy$$

$$= \alpha^* \int_Y |p(Y = y) - q(Y = y)| \, dy$$

$$= 2\alpha^* d_{TV}(p(Y), q(Y)).$$

Thus, this first property is proved and on this basis, we have the following conclusion:

$$|\mathbb{E}_T[l(h)] - \mathbb{E}_Q[l(h)]|$$

$$= \left| \int_X \int_Y l(h)t(X = x, Y = y)dxdy - \int_X \int_Y l(h)q(X = x, Y = y)dxdy \right|$$

$$\le \alpha^* \int_X \int_Y |t(X = x|Y = y)t(Y = y) - q(X = x, Y = y)q(Y = y)| \, dxdy$$

$$= \alpha^* \int_X \int_Y |t(X = x|Y = y)t(Y = y) - t(X = x|Y = y)q(Y = y)$$

$$\qquad\qquad + t(X = x|Y = y)q(Y = y) - q(X = x, Y = y)q(Y = y)|dxdy$$

$$\le \alpha^* \int_X \int_Y t(X = x|Y = y) |t(Y = y) - q(Y = y)| \, dxdy \qquad (8)$$

$$\qquad\qquad + \alpha^* \int_X \int_Y q(Y = y) |t(X = x|Y = y) - q(X = x|Y = y)| \, dxdy$$

$$= \alpha^* \int_Y |t(Y = y) - q(Y = y)| \int_X t(X = x|Y = y)dxdy$$

$$\qquad\qquad + \alpha^* \int_Y q(Y = y) \int_X |t(X = x|Y = y) - q(X = x|Y = y)|dxdy$$

$$= \alpha^* \int_Y |t(Y = y) - q(Y = y)| \, dy + 2\alpha^* \int_Y q(Y = y)d_{TV}(t(X|Y = y), q(X|Y = y))dy$$

$$= 2\alpha^* d_{TV}(t(Y), q(Y)) + 2\alpha^* \mathbb{E}_Q[d_{TV}(t(X|Y), q(X|Y))].$$

Similarly, the following formula holds

$$|\mathbb{E}_T[l(h)] - \mathbb{E}_Q[l(h)]|$$

$$\le \alpha^* \int_X \int_Y |t(X = x|Y = y)t(Y = y) - q(X = x, Y = y)q(Y = y)| \, dxdy$$

$$= \alpha^* \int_X \int_Y |t(X = x|Y = y)t(Y = y) - q(X = x|Y = y)p(Y = y) \qquad (9)$$

$$\qquad\qquad + q(X = x|Y = y)p(Y = y) - q(X = x, Y = y)q(Y = y)|dxdy$$

$$\le 2\alpha^* d_{TV}(t(Y), q(Y)) + 2\alpha^* \mathbb{E}_T[d_{TV}(t(X|Y), q(X|Y))].$$

Combining the above two formulas, we have

$$|\mathbb{E}_T[l(h)] - \mathbb{E}_Q[l(h)]| \le 2\alpha^* d_{TV}(t(Y), q(Y)) + 2\alpha^* \min\{\mathbb{E}_Q(D_{tq}(X, Y)), \mathbb{E}_T(D_{tq}(X, Y))\}, \qquad (10)$$

where $D_{tq}(X, Y) = d_{TV}(t(X|Y), q(X|Y))$.

### A.1.4 PROOF OF PROPOSITION 4

**Proposition 4** *If the conditional distribution satisfies $\hat{q}(X|Y) = p(X|Y)$, $D_{tq}(X,Y) = 0$ holds.*

**Proof:** Assume that the number of samples of class $k$ in the source set is $n_k$, we have

$$
\begin{aligned}
t(X|Y_k) &= \mathbb{E}\left(\frac{1}{n_k + \hat{m}_k}\left(\sum_{i=1}^{n_k} x_i^P + \sum_{i=1}^{\hat{m}_k} x_i^{\hat{Q}}\right)\right) \\
&= \frac{n_k}{n_k + \hat{m}_k}\mathbb{E}\left(\frac{1}{n_k}\sum_{i=1}^{n_k} x_i^P\right) + \frac{\hat{m}_k}{n_k + \hat{m}_k}\mathbb{E}\left(\frac{1}{\hat{m}_k}\sum_{i=1}^{\hat{m}_k} x_i^{\hat{Q}}\right) \\
&= \frac{n_k}{n_k + \hat{m}_k}p(X|Y_k) + \frac{\hat{m}_k}{n_k + \hat{m}_k}\hat{q}(X|Y_k) \\
&= q(X|Y_k) + \frac{\hat{m}_k}{n_k + \hat{m}_k}(\hat{q}(X|Y_k) - p(X|Y_k)), \qquad \forall k \in \{1, ..., K\}.
\end{aligned} \tag{11}
$$

Thus $t(X|Y) = q(X|Y) \Leftrightarrow \hat{q}(X|Y) = p(X|Y)$.

$$
\begin{aligned}
&D_{tq}(X,Y) \\
&= d_{TV}(t(X|Y), q(X|Y)) \\
&= \int_Y \int_X |t(X = x|Y = y) - q(X = x|Y = y)|dxdy \\
&= \int_Y \int_X |p(X = x|Y = y) + \lambda(\hat{q}(X = x|Y = y) - p(X = x|Y = y)) - q(X = x|Y = y)|dxdy \\
&= \int_Y \lambda \int_X |\hat{q}(X = x|Y = y) - p(X = x|Y = y)|dxdy.
\end{aligned} \tag{12}
$$

Through the above formula, if $\hat{q}(X|Y) = p(X|Y)$, it is obvious that $D_{tq}(X,Y) = 0$.

### A.1.5 PROOF OF PROPOSITION 5

**Proposition 5** *Assume that $\pi_i^k$ is the $i$-th of $\pi^k$, which corresponds to the weight of the $i$-th data of the $k$-th class. If $\sum_{i=1}^{\hat{m}_k} \pi_i^k = \hat{m}_k$ holds for each class $k$, then $\hat{q}_\pi(Y) = \hat{q}(Y)$ is established, where $\hat{q}_\pi(.)$ is the weighted pseudo-label distribution.*

**Proof:** We have

$$
\hat{q}(Y_k) = \frac{\sum_{i=1}^{\hat{m}} \mathbb{I}(y_i = k)}{\hat{m}} = \frac{\hat{m}_k}{\hat{m}} = \frac{\sum_{i=1}^{\hat{m}_k} \pi_i^k}{\sum_{i=1}^{\hat{m}} \sum_{k=1}^{\hat{K}} \pi_i^k} = \frac{\sum_{i=1}^{\hat{m}} \pi_i^k \mathbb{I}(y_i = k)}{\hat{m}} = \hat{q}_\pi(Y_k). \tag{13}
$$

### A.1.6 PROOF OF PROPOSITION 6

**Proposition 6** *Let the kernel mapping be universal and upper bounded that $||\varphi(x)|| \leq \mu$ for the intermediate set. Through Theorem 5, it is obvious that $||\pi^k||_1 = \hat{m}_k$. With probability at least $1 - \delta$, we have*

$$
\mathcal{D}(\hat{\pi}) - \mathcal{D}(\pi_*) \leq \varepsilon_\pi(n, \hat{m}) = 16\mu^2 \sum_{k=1}^{K} \sqrt{2\left(\frac{1}{\hat{m}_k} + \frac{1}{n_k}\right) + 2\sqrt{2\left(\frac{1}{n_k} + \frac{1}{\hat{m}_k}\right)\log\frac{K}{\delta}}}. \tag{14}
$$

**Proof:** We take the experience loss of class $k$ as $\hat{\mathcal{D}}(\pi^k; x_k^P, x_k^{\hat{Q}}) = \left\|\frac{1}{\hat{m}_k}\varphi(x_k^{\hat{Q}})\pi^k - \frac{1}{n_k}\varphi(x_k^P)\mathbf{1}\right\|^2 + \beta_k\left\|\pi^k\right\|^2$ and the expected loss of class $k$ as $\mathcal{D}(\pi^k; x_k^P, x_k^{\hat{Q}}) = \left\|\mathbb{E}\frac{1}{\hat{m}_k}\varphi(x_k^{\hat{Q}})\pi^k - \mathbb{E}\frac{1}{n_k}\varphi(x_k^P)\mathbf{1}\right\|^2 + \beta_k\left\|\pi^k\right\|^2$. Then $\hat{\mathcal{D}}(\pi) = \sum_{k=1}^{K} \hat{\mathcal{D}}(\pi^k; x_k^P, x_k^{\hat{Q}})$ and $\mathcal{D}(\pi) = \sum_{k=1}^{K} \mathcal{D}(\pi^k; x_k^P, x_k^{\hat{Q}})$ are true.

We have

$$
\left| \mathcal{D}(\hat{\pi}^k; x_k^P, x_k^{\hat{Q}}) - \mathcal{D}(\pi_*^k; x_k^P, x_k^{\hat{Q}}) \right|
$$

$$
= | \mathcal{D}(\hat{\pi}^k; x_k^P, x_k^{\hat{Q}}) - \hat{\mathcal{D}}(\hat{\pi}^k; x_k^P, x_k^{\hat{Q}}) + \hat{\mathcal{D}}(\hat{\pi}^k; x_k^P, x_k^{\hat{Q}})
$$

$$
- \hat{\mathcal{D}}(\pi_*; x_k^P, x_k^{\hat{Q}}) + \hat{\mathcal{D}}(\pi_*; x_k^P, x_k^{\hat{Q}}) - \mathcal{D}(\pi_*; x_k^P, x_k^{\hat{Q}}) | \tag{15}
$$

$$
\leq \left| \left( \mathcal{D}(\hat{\pi}^k; x_k^P, x_k^{\hat{Q}}) - \hat{\mathcal{D}}(\hat{\pi}^k; x_k^P, x_k^{\hat{Q}}) \right) + \left( \hat{\mathcal{D}}(\pi_*; x_k^P, x_k^{\hat{Q}}) - \mathcal{D}(\pi_*; x_k^P, x_k^{\hat{Q}}) \right) \right|
$$

$$
\leq 2 \sup_{\pi^k} \left| \mathcal{D}(\pi^k; x_k^P, x_k^{\hat{Q}}) - \hat{\mathcal{D}}(\pi^k; x_k^P, x_k^{\hat{Q}}) \right|,
$$

where the first inequality holds because $\hat{\pi}^k$ is the empirical minimizer of $\hat{\mathcal{D}}(\pi^k; x_k^P, x_k^{\hat{Q}})$ and thus $\hat{\mathcal{D}}(\hat{\pi}^k; x_k^P, x_k^{\hat{Q}}) - \hat{\mathcal{D}}(\pi_*; x_k^P, x_k^{\hat{Q}}) \leq 0$.

Further, we have

$$
\left| \mathcal{D}(\pi^k; x_k^P, x_k^{\hat{Q}}) - \hat{\mathcal{D}}(\pi^k; x_k^P, x_k^{\hat{Q}}) \right|
$$

$$
= \left| \left\| \frac{1}{\hat{m}_k} \varphi(x_k^{\hat{Q}}) \pi^k - \frac{1}{n_k} \varphi(x_k^P) \mathbf{1} \right\|^2 + \beta_k \|\pi^k\|^2 - \left\| \mathbb{E} \frac{1}{\hat{m}_k} \varphi(x_k^{\hat{Q}}) \pi^k - \mathbb{E} \frac{1}{n_k} \varphi(x_k^P) \mathbf{1} \right\|^2 - \beta_k \|\pi^k\|^2 \right|
$$

$$
= \left| \left\| \frac{1}{\hat{m}_k} \varphi(x_k^{\hat{Q}}) \pi^k - \frac{1}{n_k} \varphi(x_k^P) \mathbf{1} \right\|^2 - \left\| \mathbb{E} \frac{1}{\hat{m}_k} \varphi(x_k^{\hat{Q}}) \pi^k - \mathbb{E} \frac{1}{n_k} \varphi(x_k^P) \mathbf{1} \right\|^2 \right|
$$

$$
= \left( \mathbb{E} \left( \frac{1}{\hat{m}_k} \varphi(x_k^{\hat{Q}}) \pi^k - \frac{1}{n_k} \varphi(x_k^P) \mathbf{1} \right) - \left( \frac{1}{\hat{m}_k} \varphi(x_k^{\hat{Q}}) \pi^k - \frac{1}{n_k} \varphi(x_k^P) \mathbf{1} \right) \right)^T
$$

$$
\left( \mathbb{E} \left( \frac{1}{\hat{m}_k} \varphi(x_k^{\hat{Q}}) \pi^k - \frac{1}{n_k} \varphi(x_k^P) \mathbf{1} \right) + \left( \frac{1}{\hat{m}_k} \varphi(x_k^{\hat{Q}}) \pi^k - \frac{1}{n_k} \varphi(x_k^P) \mathbf{1} \right) \right)
$$

$$
\leq \left\| \mathbb{E} \left( \frac{1}{\hat{m}_k} \varphi(x_k^{\hat{Q}}) \pi^k - \frac{1}{n_k} \varphi(x_k^P) \mathbf{1} \right) - \left( \frac{1}{\hat{m}_k} \varphi(x_k^{\hat{Q}}) \pi^k - \frac{1}{n_k} \varphi(x_k^P) \mathbf{1} \right) \right\|
$$

$$
\left\| \mathbb{E} \left( \frac{1}{\hat{m}_k} \varphi(x_k^{\hat{Q}}) \pi^k - \frac{1}{n_k} \varphi(x_k^P) \mathbf{1} \right) + \left( \frac{1}{\hat{m}_k} \varphi(x_k^{\hat{Q}}) \pi^k - \frac{1}{n_k} \varphi(x_k^P) \mathbf{1} \right) \right\|. \tag{16}
$$

Due to $\|\varphi(x)\| \leq \mu$ and $\|\pi^k\|_1 = \hat{m}_k$, then

$$
\left\| \mathbb{E} \left( \frac{1}{\hat{m}_k} \varphi(x_k^{\hat{Q}}) \pi^k - \frac{1}{n_k} \varphi(x_k^P) \mathbf{1} \right) + \left( \frac{1}{\hat{m}_k} \varphi(x_k^{\hat{Q}}) \pi^k - \frac{1}{n_k} \varphi(x_k^P) \mathbf{1} \right) \right\|
$$

$$
\leq \left\| \mathbb{E} \frac{1}{\hat{m}_k} \varphi(x_k^{\hat{Q}}) \pi^k \right\| + \left\| \mathbb{E} \frac{1}{n_k} \varphi(x_k^P) \mathbf{1} \right\| + \left\| \frac{1}{\hat{m}_k} \varphi(x_k^{\hat{Q}}) \pi^k \right\| + \left\| \frac{1}{n_k} \varphi(x_k^P) \mathbf{1} \right\| \tag{17}
$$

$$
\leq \mu + \mu + \mu + \mu
$$

$$
= 4\mu.
$$

Let $f(x_k^{\hat{Q}}, x_k^P, \pi^k) \triangleq \mathbb{E} \left( \frac{1}{\hat{m}_k} \varphi(x_k^{\hat{Q}}) \pi^k - \frac{1}{n_k} \varphi(x_k^P) \mathbf{1} \right) - \left( \frac{1}{\hat{m}_k} \varphi(x_k^{\hat{Q}}) \pi^k - \frac{1}{n_k} \varphi(x_k^P) \mathbf{1} \right)$. Combining the above two equations, we have

$$
\left| \mathcal{D}(\pi^k; x_k^P, x_k^{\hat{Q}}) - \hat{\mathcal{D}}(\pi^k; x_k^P, x_k^{\hat{Q}}) \right| \leq 4\mu \left\| f(x_k^{\hat{Q}}, x_k^P, \pi^k) \right\|. \tag{18}
$$

Combining Eq. (15) and Eq. (18), we have

$$
\mathcal{D}(\hat{\pi}^k; x_k^P, x_k^{\hat{Q}}) - \mathcal{D}(\pi_*^k; x_k^P, x_k^{\hat{Q}}) \leq 2 \sup_{\pi^k} \left| \mathcal{D}(\pi^k; x_k^P, x_k^{\hat{Q}}) - \hat{\mathcal{D}}(\pi^k; x_k^P, x_k^{\hat{Q}}) \right|
$$

$$
\leq 8\mu \sup_{\pi^k} \left\| f(x_k^{\hat{Q}}, x_k^P, \pi^k) \right\|. \tag{19}
$$

It is hard to upper bound the $\sup_{\pi^k} \left\| f(x_k^{\hat{Q}}, x_k^P, \pi^k) \right\|$ directly and let's start with the expectation upper bound, i.e., $\mathbb{E} \sup_{\pi^k} \left\| f(x_k^{\hat{Q}}, x_k^P, \pi^k) \right\|$, in the first. We further define a ghost data set $[\bar{x}_k^P, \bar{x}_k^Q]_{k=1}^K$ and

the corresponding ghost loss is

$$
\mathbb{E}\sup_{\pi^k}\left\|f(x_k^{\hat{Q}}, x_k^P, \pi^k)\right\|
$$

$$
= \mathbb{E}\sup_{\pi^k}\left\|\mathbb{E}\left(\frac{1}{\hat{m}_k}\varphi(x_k^{\hat{Q}})\pi^k - \frac{1}{n_k}\varphi(x_k^P)\mathbf{1}\right) - \left(\frac{1}{\hat{m}_k}\varphi(x_k^{\hat{Q}})\pi^k - \frac{1}{n_k}\varphi(x_k^P)\mathbf{1}\right)\right\|
$$

$$
= \mathbb{E}_{x_k^P, x_k^{\hat{Q}}}\sup_{\pi^k}\left\|\mathbb{E}_{\bar{x}_k^P, \bar{x}_k^{\hat{Q}}}\left(\frac{1}{\hat{m}_k}\varphi(\bar{x}_k^{\hat{Q}})\pi^k - \frac{1}{n_k}\varphi(\bar{x}_k^P)\mathbf{1}\right) - \left(\frac{1}{\hat{m}_k}\varphi(x_k^{\hat{Q}})\pi^k - \frac{1}{n_k}\varphi(x_k^P)\mathbf{1}\right)\right\| \quad (20)
$$

$$
\leq \mathbb{E}_{x_k^P, x_k^{\hat{Q}}, \bar{x}_k^P, \bar{x}_k^{\hat{Q}}}\sup_{\pi^k}\left\|\left(\frac{1}{\hat{m}_k}\varphi(\bar{x}_k^{\hat{Q}})\pi^k - \frac{1}{n_k}\varphi(\bar{x}_k^P)\mathbf{1}\right) - \left(\frac{1}{\hat{m}_k}\varphi(x_k^{\hat{Q}})\pi^k - \frac{1}{n_k}\varphi(x_k^P)\mathbf{1}\right)\right\|.
$$

Due to Jensens Inequality and the convexity of $L_2$ norm, the last inequality holds.

Since the random variable $\{\frac{1}{\hat{m}_k}\varphi(\bar{x}_k^{\hat{Q}})\pi^k - \frac{1}{n_k}\varphi(\bar{x}_k^P)\mathbf{1} - \frac{1}{\hat{m}_k}\varphi(x_k^{\hat{Q}})\pi^k + \frac{1}{n_k}\varphi(x_k^P)\mathbf{1}\}$ is a symmetric random variable, we use the usual symmetrizing technique through Rademacher variables $\{\xi_i\}_{i=1}^n$ which are uniformly distributed from $\{1, -1\}$. Then we have

$$
\begin{cases}
\varphi(x_k^P, \xi) \triangleq [\xi_1\varphi(x_{k,1}^P), \xi_2\varphi(x_{k,2}^P), ..., \xi_n\varphi(x_{k,n_k}^P)]^T, \\
\varphi(x_k^{\hat{Q}}, \xi) \triangleq [\xi_1\varphi(x_{k,1}^{\hat{Q}}), \xi_2\varphi(x_{k,2}^{\hat{Q}}), ..., \xi_n\varphi(x_{k,\hat{m}_k}^{\hat{Q}})]^T.
\end{cases} \quad (21)
$$

By the above definition, it's obvious that the random variable $\{\frac{1}{\hat{m}_k}\varphi(\bar{x}_k^{\hat{Q}}, \xi)\pi^k - \frac{1}{n_k}\varphi(\bar{x}_k^P, \xi) - \frac{1}{\hat{m}_k}\varphi(x_k^{\hat{Q}}, \xi)\pi^k + \frac{1}{n_k}\varphi(x_k^P, \xi)\}$ has the same distribution as $\{\frac{1}{\hat{m}_k}\varphi(\bar{x}_k^{\hat{Q}})\pi^k - \frac{1}{n_k}\varphi(\bar{x}_k^P) - \frac{1}{\hat{m}_k}\varphi(x_k^{\hat{Q}})\pi^k + \frac{1}{n_k}\varphi(x_k^P)\}$. Then we have

$$
\mathbb{E}_{x_k^P, x_k^{\hat{Q}}, \bar{x}_k^P, \bar{x}_k^{\hat{Q}}}\sup_{\pi^k}\left\|\left(\frac{1}{\hat{m}_k}\varphi(\bar{x}_k^{\hat{Q}})\pi^k - \frac{1}{n_k}\varphi(\bar{x}_k^P)\mathbf{1}\right) - \left(\frac{1}{\hat{m}_k}\varphi(x_k^{\hat{Q}})\pi^k - \frac{1}{n_k}\varphi(x_k^P)\mathbf{1}\right)\right\|
$$

$$
= \mathbb{E}_{x_k^P, x_k^{\hat{Q}}, \bar{x}_k^P, \bar{x}_k^{\hat{Q}}, \xi}\sup_{\pi^k}\left\|\left(\frac{1}{\hat{m}_k}\varphi(\bar{x}_k^{\hat{Q}}, \xi)\pi^k - \frac{1}{n_k}\varphi(\bar{x}_k^P, \xi)\mathbf{1}\right) - \left(\frac{1}{\hat{m}_k}\varphi(x_k^{\hat{Q}}, \xi)\pi^k - \frac{1}{n_k}\varphi(x_k^P, \xi)\mathbf{1}\right)\right\|
$$

$$
\leq 2\mathbb{E}_{x_k^P, x_k^{\hat{Q}}, \xi}\sup_{\pi^k}\left\|\frac{1}{\hat{m}_k}\varphi(x_k^{\hat{Q}}, \xi)\pi^k - \frac{1}{n_k}\varphi(x_k^P, \xi)\mathbf{1}\right\|
$$

$$
\leq 2\mathbb{E}_{x_k^{\hat{Q}}, \xi}\sup_{\pi^k}\left\|\frac{1}{\hat{m}_k}\varphi(x_k^{\hat{Q}}, \xi)\pi^k\right\| + 2\mathbb{E}_{x_k^P, \xi}\sup_{\pi^k}\left\|\frac{1}{n_k}\varphi(x_k^P, \xi)\mathbf{1}\right\|,
$$

$$
(22)
$$

where the inequalities hold because of the triangle inequality. In the next proof, let's look at the upper bound of $\mathbb{E}_{x_k^{\hat{Q}}, \xi}\sup_{\pi^k}\left\|\frac{1}{\hat{m}_k}\varphi(x_k^{\hat{Q}}, \xi)\pi^k\right\|$ and $\mathbb{E}_{x_k^P, \xi}\sup_{\pi^k}\left\|\frac{1}{n_k}\varphi(x_k^P, \xi)\mathbf{1}\right\|$, respectively.

$$
\mathbb{E}_{x_k^{\hat{Q}}, \xi}\sup_{\pi^k}\left\|\frac{1}{\hat{m}_k}\varphi(x_k^{\hat{Q}}, \xi)\pi^k\right\| = \mathbb{E}_{x_k^{\hat{Q}}, \xi}\sup_{\pi^k}\left\|\frac{1}{\hat{m}_k}[\xi_1\varphi(x_{k,1}^{\hat{Q}}), \xi_2\varphi(x_{k,2}^{\hat{Q}}), ..., \xi_n\varphi(x_{k,\hat{m}_k}^{\hat{Q}})]^T\pi^k\right\|
$$

$$
= \frac{1}{\hat{m}_k}\mathbb{E}_{x_k^{\hat{Q}}, \xi}\sup_{\pi^k}\left\|[\xi_1\varphi(x_{k,1}^{\hat{Q}}), \xi_2\varphi(x_{k,2}^{\hat{Q}}), ..., \xi_n\varphi(x_{k,\hat{m}_k}^{\hat{Q}})]^T\pi^k\right\|
$$

$$
\leq \frac{1}{\hat{m}_k}\mu\mathbb{E}_{\xi}\sup_{\pi^k}\sqrt{\sum_{i=1}^{\hat{m}_k}\xi_i^2\pi_i^k}
$$

$$
\leq \frac{1}{\hat{m}_k}\mu\sup_{\pi^k}\sqrt{\sum_{i=1}^{\hat{m}_k}\pi_i^k}
$$

$$
= \frac{1}{\hat{m}_k}\mu\sqrt{\|\pi^k\|_1}
$$

$$
= \frac{\mu}{\sqrt{\hat{m}_k}}.
$$

$$
(23)
$$

The first inequality holds because of the Cauchy-Schwarz inequality and the second inequality holds on account of the Talagrand Contraction Lemma Ledoux & Talagrand (1991). Similarly, the following conclusion holds true.

$$\mathbb{E}_{x_k^P, \xi} \sup_{\pi^k} \left\| \frac{1}{n_k} \varphi(x_k^P, \xi) \mathbf{1} \right\| \leq \frac{\mu}{\sqrt{n_k}}. \tag{24}$$

Combining Eq. (20), Eq. (22), Eq. (23) and Eq. (24), we have

$$\mathbb{E} \sup_{\pi^k} \left\| f(x_k^{\hat{Q}}, x_k^P, \pi^k) \right\| \leq 2\mu \left( \frac{1}{\sqrt{\hat{m}_k}} + \frac{1}{\sqrt{n_k}} \right). \tag{25}$$

Assume that the $i$-th variable of $x_k^{\hat{Q}}$ is replaced by a new independent variable $\tilde{x}_{k,i}^{\hat{Q}}$ and the new example is $x_k^{\hat{Q}_i}$, where $i \in \{1, 2, ..., \hat{m}_k\}$ and the $j$-th variable of $x_k^P$ is replaced by a new independent variable $\tilde{x}_{k,j}^P$ and the new example is $x_k^{P_j}$, where $j \in \{1, 2, ..., n_k\}$. Then for any $i \in \{1, 2, ..., \hat{m}_k\}$, we have

$$\left| \sup_{\pi^k} \left\| f(x_k^{\hat{Q}_i}, x_k^P, \pi^k) \right\|^2 - \sup_{\pi^k} \left\| f(x_k^{\hat{Q}}, x_k^P, \pi^k) \right\|^2 \right|$$

$$\leq \sup_{\pi^k} \left| \left( f(x_k^{\hat{Q}_i}, x_k^P, \pi^k) + f(x_k^{\hat{Q}}, x_k^P, \pi^k) \right)^T \left( f(x_k^{\hat{Q}_i}, x_k^P, \pi^k) - f(x_k^{\hat{Q}}, x_k^P, \pi^k) \right) \right|$$

$$\leq \left| 8\varphi_{\max}^T \left( f(x_k^{\hat{Q}_i}, x_k^P, \pi^k) - f(x_k^{\hat{Q}}, x_k^P, \pi^k) \right) \right|$$

$$= \left| 8\varphi_{\max}^T \left( \frac{1}{\hat{m}_k} \varphi(x_k^{\hat{Q}_i}) \pi^k - \frac{1}{\hat{m}_k} \varphi(x_k^{\hat{Q}}) \pi^k \right) \right| \tag{26}$$

$$\leq \frac{16}{\hat{m}_k} \varphi_{\max}^T \varphi_{\max}$$

$$= \frac{16\mu^2}{\hat{m}_k}.$$

Similarly, for any $j \in \{1, 2, ..., n_k\}$, the following conclusion holds true.

$$\left| \sup_{\pi^k} \left\| f(x_k^{\hat{Q}}, x_k^{P_j}, \pi^k) \right\|^2 - \sup_{\pi^k} \left\| f(x_k^{\hat{Q}}, x_k^P, \pi^k) \right\|^2 \right|$$

$$\leq \sup_{\pi^k} \left| \left( f(x_k^{\hat{Q}}, x_k^{P_j}, \pi^k) + f(x_k^{\hat{Q}}, x_k^P, \pi^k) \right)^T \left( f(x_k^{\hat{Q}}, x_k^{P_j}, \pi^k) - f(x_k^{\hat{Q}}, x_k^P, \pi^k) \right) \right| \tag{27}$$

$$= \left| 8\varphi_{\max}^T \left( \frac{1}{n_k} \varphi(x_k^{P_j}) \mathbf{1} - \frac{1}{n_k} \varphi(x_k^P) \mathbf{1} \right) \right|$$

$$= \frac{16\mu^2}{n_k}.$$

By employing the McDiarmids Inequality , we have

$$P \left[ \sup_{\pi^k} \left\| f(x_k^{\hat{Q}}, x_k^P, \pi^k) \right\|^2 - \mathbb{E}_{\{x_k^{\hat{Q}}, x_k^P\}} \sup_{\pi^k} \left\| f(x_k^{\hat{Q}}, x_k^P, \pi^k) \right\|^2 \geq \varepsilon \right] \leq \exp \left( \frac{-2\varepsilon^2}{256\mu^4 \left( \frac{1}{n_k} + \frac{1}{\hat{m}_k} \right)} \right). \tag{28}$$

Let

$$\delta = \exp \left( \frac{-2\varepsilon^2}{256\mu^4 \left( \frac{1}{n_k} + \frac{1}{\hat{m}_k} \right)} \right), \tag{29}$$

and for any $\delta \geq 0$, with probability at least $1 - \delta$, we have

$$\sup_{\pi^k} \left\| f(x_k^{\hat{Q}}, x_k^P, \pi^k) \right\|^2 - \mathbb{E}_{\{x_k^{\hat{Q}}, x_k^P\}} \sup_{\pi^k} \left\| f(x_k^{\hat{Q}}, x_k^P, \pi^k) \right\|^2 \leq 8\mu^2 \sqrt{2 \left( \frac{1}{n_k} + \frac{1}{\hat{m}_k} \right) \log \frac{1}{\delta}}. \tag{30}$$

Due to $\mathbb{E}_{\{x_k^{\hat{Q}}, x_k^P\}} \sup_{\pi^k} \left\| f(x_k^{\hat{Q}}, x_k^P, \pi^k) \right\| \le 2\mu \left( \frac{1}{\sqrt{\hat{m}_k}} + \frac{1}{\sqrt{n_k}} \right)$ mentioned in Eq. (27), we have

$$
\begin{aligned}
\mathbb{E}_{\{x_k^{\hat{Q}}, x_k^P\}} \sup_{\pi^k} \left\| f(x_k^{\hat{Q}}, x_k^P, \pi^k) \right\|^2 &\le 4\mu^2 \left( \frac{1}{\sqrt{\hat{m}_k}} + \frac{1}{\sqrt{n_k}} \right)^2 \\
&\le 8\mu^2 \left( \frac{1}{\hat{m}_k} + \frac{1}{n_k} \right).
\end{aligned}
\tag{31}
$$

Thus

$$
\begin{aligned}
\sup_{\pi^k} \left\| f(x_k^{\hat{Q}}, x_k^P, \pi^k) \right\| &\le \sqrt{ \mathbb{E}_{\{x_k^{\hat{Q}}, x_k^P\}} \sup_{\pi^k} \left\| f(x_k^{\hat{Q}}, x_k^P, \pi^k) \right\|^2 + 8\mu^2 \sqrt{2 \left( \frac{1}{n_k} + \frac{1}{\hat{m}_k} \right) \log \frac{1}{\delta}} } \\
&\le \sqrt{ 8\mu^2 \left( \frac{1}{\hat{m}_k} + \frac{1}{n_k} \right) + 8\mu^2 \sqrt{2 \left( \frac{1}{n_k} + \frac{1}{\hat{m}_k} \right) \log \frac{1}{\delta}} } \\
&= 2\mu \sqrt{ 2 \left( \frac{1}{\hat{m}_k} + \frac{1}{n_k} \right) + 2 \sqrt{2 \left( \frac{1}{n_k} + \frac{1}{\hat{m}_k} \right) \log \frac{1}{\delta}} },
\end{aligned}
\tag{32}
$$

and

$$
\begin{aligned}
\mathcal{D}(\hat{\pi}^k; x_k^P, x_k^{\hat{Q}}) - \mathcal{D}(\pi_*^k; x_k^P, x_k^{\hat{Q}}) &\le 8\mu \sup_{\pi^k} \left\| f(x_k^{\hat{Q}}, x_k^P, \pi^k) \right\| \\
&\le 16\mu^2 \sqrt{ 2 \left( \frac{1}{\hat{m}_k} + \frac{1}{n_k} \right) + 2 \sqrt{2 \left( \frac{1}{n_k} + \frac{1}{\hat{m}_k} \right) \log \frac{1}{\delta}} }.
\end{aligned}
\tag{33}
$$

Due to $\mathcal{D}(\pi) = \sum_{k=1}^{K} \mathcal{D}(\pi^k; x_k^P, x_k^{\hat{Q}})$, with probability at least $1 - \delta$, we have

$$
\mathcal{D}(\hat{\pi}) - \mathcal{D}(\pi_*) \le \varepsilon_\pi(n, \hat{m}) \le \sum_{k=1}^{K} 16\mu^2 \sqrt{ 2 \left( \frac{1}{\hat{m}_k} + \frac{1}{n_k} \right) + 2 \sqrt{2 \left( \frac{1}{n_k} + \frac{1}{\hat{m}_k} \right) \log \frac{K}{\delta}} }.
\tag{34}
$$

### A.1.7 PROOF OF PROPOSITION 7

**Proposition 7** *We assume that the true importance weight is $\tau(X, Y) := \frac{q(X,Y)}{t(X,Y)}$ and the pseudo importance weight is $w_t(Y)$. In the case of ALS framework based on RLLS, with probability at least $1 - 3\delta$, the ALS generalizes as:*

$$
\begin{aligned}
\mathcal{L}(\hat{h}_{\hat{w}_t}; \tau) - \mathcal{L}(h^*; \tau) \le{}& \varepsilon_{\mathcal{G}}(n + \hat{m}, d_\infty(q||t), d(q||t)) + \varepsilon_w(n + \hat{m}, \|w - 1\|_2) \\
&+ \frac{2\alpha^* rm}{n + rm} \left( d_{X|Y}(q||\hat{q}) + d_Y(q||\hat{q}) \right) + 2\alpha^* \|w_t(Y)\|_\infty \, err(h_{\tilde{v}}).
\end{aligned}
\tag{35}
$$

*Where $\varepsilon_{\mathcal{G}}(n + \hat{m}, d_\infty(q||t), d(q||t)) = 2\mathcal{R}(\mathcal{G}(\mathcal{L}, \mathcal{H})) + \frac{2\alpha^* d_\infty(q||t) \log(2/\delta)}{n + \hat{m}} + \sqrt{\frac{2\alpha^* d(q||t) \log(2/\delta)}{n + \hat{m}}}$, $\varepsilon_w(n + \hat{m}, \|w - 1\|_2) = 2\lambda\alpha^* \mathcal{O}\left( \frac{1}{\sigma_{\min}} \left( \|w - 1\|_2 \sqrt{\frac{\log(K/\delta)}{n + \hat{m}}} + \sqrt{\frac{\log(1/\delta)}{n + \hat{m}}} + \sqrt{\frac{\log(1/\delta)}{m}} \right) \right) + 2\alpha^*(1 - \lambda)\|w - 1\|_2$, $err(h_{\tilde{v}})$ denotes the importance weighted 0/1-error of the predictor $h_{\tilde{v}}$.*

**Proof:** Let's review the definition and representation of the following equations:

$$
\begin{cases}
\mathcal{L}(h; \tau) = \mathbb{E}_{X,Y \in T}[\tau(Y) l(h(X), Y)], \\
\mathcal{L}(h; w_t) = \mathbb{E}_{X,Y \in T}[w_t(\hat{Y}) l(h(X), \hat{Y})], \\
\hat{\mathcal{L}}(h; w_t) = \frac{1}{n + \hat{m}} \sum_{i=1}^{n + \hat{m}} w_t(\hat{y}_i) \hat{\pi}(x_i) l(h(x_i), \hat{y}_i).
\end{cases}
\tag{36}
$$

By addition and subtraction we have

$$
\begin{aligned}
&\mathcal{L}(\hat{h}_{\hat{w}_t}; \tau) - \mathcal{L}(h^*; \tau) \\
&= \left( \mathcal{L}(\hat{h}_{\hat{w}_t}; \tau) - \mathcal{L}(\hat{h}_{\hat{w}_t}; w_t) \right) + \left( \mathcal{L}(\hat{h}_{\hat{w}_t}; w_t) - \hat{\mathcal{L}}(\hat{h}_{\hat{w}_t}; w_t) \right) + \left( \hat{\mathcal{L}}(\hat{h}_{\hat{w}_t}; w_t) - \hat{\mathcal{L}}(\hat{h}_{\hat{w}_t}; \hat{w}_t) \right) \\
&\qquad + \left( \hat{\mathcal{L}}(\hat{h}_{\hat{w}_t}; \hat{w}_t) - \hat{\mathcal{L}}(h^*; \hat{w}_t) \right) + \left( \hat{\mathcal{L}}(h^*; \hat{w}_t) - \hat{\mathcal{L}}(h^*; w_t) \right) \\
&\qquad + \left( \hat{\mathcal{L}}(h^*; w_t) - \mathcal{L}(h^*; w_t) \right) + \left( \mathcal{L}(h^*; w_t) - \mathcal{L}(h^*; \tau) \right) \\
&\leq 2 \sup_{h \in \mathcal{H}} \left( \mathcal{L}(h; \tau) - \mathcal{L}(h; w_t) \right) + 2 \sup_{h \in \mathcal{H}} \left( \mathcal{L}(h; w_t) - \hat{\mathcal{L}}(h; w_t) \right) + 2 \sup_{h \in \mathcal{H}} \left( \hat{\mathcal{L}}(h; w_t) - \hat{\mathcal{L}}(h; \hat{w}_t) \right),
\end{aligned}
$$
(37)

where the first inequality holds because $\hat{h}_{\hat{w}_t}$ is the empirical minimizer of $\hat{\mathcal{L}}(h; \hat{w}_t)$ and thus $\hat{\mathcal{L}}(\hat{h}_{\hat{w}_t}; \hat{w}_t) - \hat{\mathcal{L}}(h^*; \hat{w}_t) \leq 0$. Now let's analyze the upper bound of each part.

[1] We analyze the upper bound of $\mathcal{L}(h; \tau) - \mathcal{L}(h; w_t)$ in the first.

$$
\begin{aligned}
&\mathcal{L}(h; \tau) - \mathcal{L}(h; w_t) \\
&= \mathbb{E}_{X,Y \in T} \left[ \tau(Y) l(h(X), Y) \right] - \mathbb{E}_{X,Y \in T} \left[ w_t(\hat{Y}) l(h(X), \hat{Y}) \right] \\
&= \mathbb{E}_{X,Y \in T} \left[ \tau(Y) l(h(X), Y) \right] - \mathbb{E}_{X,Y \in T} \left[ w_t(\hat{Y}) l(h(X), Y) \right] \\
&\qquad\qquad\qquad + \mathbb{E}_{X,Y \in T} \left[ w_t(\hat{Y}) l(h(X), Y) \right] - \mathbb{E}_{X,Y \in T} \left[ w_t(\hat{Y}) l(h(X), \hat{Y}) \right] \\
&= \mathbb{E}_{X,Y \in T} \left[ \left( \tau(Y) - w_t(\hat{Y}) \right) l(h(X), Y) \right] + \mathbb{E}_{X,Y \in T} \left[ w_t(\hat{Y}) \left( l(h(X), Y) - l(h(X), \hat{Y}) \right) \right] \\
&= \mathbb{E}_{X,Y \in T} \left[ \left( \tau(Y) - w_*(Y) \right) l(h(X), Y) \right] + \mathbb{E}_{X,Y \in T} \left[ \left( w_*(Y) - w_t(\hat{Y}) \right) l(h(X), Y) \right] \\
&\qquad\qquad\qquad + \mathbb{E}_{X,Y \in T} \left[ w_t(\hat{Y}) \left( l(h(X), Y) - l(h(X), \hat{Y}) \right) \right].
\end{aligned}
$$
(38)

Where $\mathbb{E}_{X,Y \in T} \left[ (\tau(Y) - w_*(Y)) l(h(X), Y) \right]$ represents the presence of $\hat{\pi}$ which leads to the deviation from the homogeneous conditional distribution assumption.

$$
\begin{aligned}
&\mathbb{E}_{X,Y \in T} \left[ (\tau(Y) - w_*(Y)) l(h(X), Y) \right] \\
&= \mathbb{E}_{X,Y \in T} \left[ \left( \frac{q(X|Y)}{t(X|Y)} - 1 \right) w_*(Y) l(h(X), Y) \right] \\
&= \mathbb{E}_{X,Y \in Q} \left[ \left( \frac{q(X|Y)}{t(X|Y)} - 1 \right) l(h(X), Y) \right] \\
&\leq \alpha^* \mathbb{E}_{X,Y \in Q} \left[ \left| \frac{q(X|Y)}{t(X|Y)} - 1 \right| \right] \\
&= \alpha^* \mathbb{E}_{X,Y \in Q} \left[ \frac{1}{t(X|Y)} |q(X|Y) - t(X|Y)| \right]
\end{aligned}
$$
(39)

In addition, we have $t(X|Y) = \lambda p(X|Y) + (1-\lambda)\hat{q}(X|Y)$, where $\lambda = \frac{n}{n+rm}$. Thus,

$$
\begin{aligned}
&\mathbb{E}_{X,Y \in T} \left[ (\tau(Y) - w_*(Y)) l(h(X), Y) \right] \\
&\leq \alpha^* \mathbb{E}_{X,Y \in Q} \left[ \frac{1}{t(X|Y)} |q(X|Y) - t(X|Y)| \right] \\
&= \frac{\alpha^* rm}{n + rm} \mathbb{E}_{X,Y \in Q} \left[ \frac{1}{t(X|Y)} |q(X|Y) - \hat{q}(X|Y)| \right]
\end{aligned}
$$
(40)

Then we analyse the bound of $\mathbb{E}_{X,Y \in T}\left[(w_*(Y) - w_t(Y))\, l(h(X), Y)\right]$, which is caused by the difference between the true and pseudo label weight.

$$
\mathbb{E}_{X,Y \in T}\left[\left(w_*(Y) - w_t(\hat{Y})\right) l(h(X), Y)\right]
$$

$$
= \mathbb{E}_{X,Y \in T}\left[w_*(Y)\left(1 - \frac{t(Y)}{t(\hat{Y})}\right) l(h(X), Y)\right]
$$

$$
\leq \alpha^* \mathbb{E}_{X,Y \in Q}\left[\left|1 - \frac{t(Y)}{t(\hat{Y})}\right|\right] \tag{41}
$$

$$
= \alpha^* \mathbb{E}_{X,Y \in Q}\left[\frac{1}{t(\hat{Y})}\left|t(\hat{Y}) - t(Y)\right|\right]
$$

$$
= \frac{\alpha^* rm}{n + rm} \mathbb{E}_{X,Y \in Q}\left[\frac{1}{t(\hat{Y})}\left|q(Y) - \hat{q}(Y)\right|\right]
$$

Finally, we analyse the bound of $\mathbb{E}_{X,Y \in T}\left[w_t(\hat{Y})\left(l(h(X), Y) - l(h(X), \hat{Y})\right)\right]$, which is caused by the prediction accuracy.

$$
\mathbb{E}_{X,Y \in T}\left[w_t(\hat{Y})\left(l(h(X), Y) - l(h(X), \hat{Y})\right)\right]
$$

$$
\leq \alpha^* \mathbb{E}_{X,Y \in T}\left[w_t(\hat{Y})\mathbb{I}(\hat{Y} \neq Y)\right]
$$

$$
\leq \alpha^* \left\|w_t(\hat{Y})\right\|_\infty \mathbb{E}_{X,Y \in T}\left[\mathbb{I}(\hat{Y} \neq Y)\right] \tag{42}
$$

$$
= \alpha^* \left\|w_t(\hat{Y})\right\|_\infty \mathrm{err}(h_{\tilde{v}}),
$$

where $\mathrm{err}(h_{\tilde{v}})$ denotes the importance weighted 0/1-error of the predictor $h_{\tilde{v}}$. Combining the eq. (40), eq. (41) and eq. (42), we have

$$
\mathcal{L}(h; \tau) - \mathcal{L}(h; w_t)
$$

$$
\leq \frac{\alpha^* rm}{n + rm}\left(\mathbb{E}_{X,Y \in Q}\left[\frac{1}{t(X|Y)}\left|q(X|Y) - \hat{q}(X|Y)\right|\right] + \mathbb{E}_{X,Y \in Q}\left[\frac{1}{t(\hat{Y})}\left|q(Y) - \hat{q}(Y)\right|\right]\right)
$$

$$
+ \alpha^* \left\|w_t(Y)\right\|_\infty \mathbb{E}_{X,Y \in T}\left[\mathbb{I}(\hat{Y} \neq Y)\right] \tag{43}
$$

[2] Then we analyze the upper bound of $\mathcal{L}(h; w_t) - \hat{\mathcal{L}}(h; w_t)$.

Let $[\bar{x}_k^P, \bar{x}_k^Q]_{k=1}^K$ be the ghost samples which are i.i.d. copies of $[x_k^P, x_k^Q]_{k=1}^K$ and the corresponding ghost loss is

$$
\hat{\mathcal{L}}'(h; w_t) = \frac{1}{n + \hat{m}} \sum_{i=1}^{n+\hat{m}} w_t(\bar{y}_i^T)\hat{\pi}(\bar{x}_i^T) l(h(\bar{x}_i^T), \bar{y}_i^T). \tag{44}
$$

Let's define a random variable $G_{n+\hat{m}} := \hat{\mathcal{L}}(h; w_t) - \mathcal{L}(h; w_t)$, and then

$$
\mathbb{E}\left[G_{n+\hat{m}}\right] = \mathbb{E}\left[\sup_{h \in \mathcal{H}} \hat{\mathcal{L}}(h; w_t) - \mathbb{E}\left[\hat{\mathcal{L}}'(h; w_t)\right]\right]
$$

$$
\leq \mathbb{E}\left[\mathbb{E}\left[\sup_{h \in \mathcal{H}} \hat{\mathcal{L}}(h; w_t) - \hat{\mathcal{L}}'(h; w_t) \,\middle|\, \{(x_i^T, y_i^T)\}_1^{n+\hat{m}}\right]\right]. \tag{45}
$$

$$
\leq 2\mathbb{E}\left[\sup_{h \in \mathcal{H}} \frac{1}{n + \hat{m}} \sum_{i=1}^{n+\hat{m}} w_t(y_i^T)\hat{\pi}(x_i^T) l(h(x_i^T), y_i^T)\right]
$$

$$
= 2\mathcal{R}_{n+\hat{m}}(\mathcal{G}(\mathcal{L}, \mathcal{H})).
$$

Where $\mathcal{R}_{n+\hat{m}}(\mathcal{G}(\mathcal{L}, \mathcal{H})) := \mathbb{E}_{(x_i^T, y_i^T) \in T}\left[\mathbb{E}_{\xi_i} \frac{1}{n+\hat{m}}\left[\sup_{h \in \mathcal{H}} \sum_{i=1}^{n+\hat{m}} \xi_i \hat{\pi}(x_i^T) g_h(x_i^T, y_i^T)\right]\right]$. Consider a sequence of Doob Martingale and filtration $(\mathcal{U}_j, \mathcal{F}_j)$ Azizzadenesheli et al. (2019) :

$$\mathcal{U}_j := \mathbb{E}\left[\sup_{h \in \mathcal{H}} \frac{1}{n+\hat{m}} \sum_{i=1}^{n+\hat{m}} w_t(y_i^T) \hat{\pi}(x_i^T) l(h(x_i^T), y_i^T) \Big| \{(x_i^T, y_i^T)\}_1^j\right], \qquad (46)$$

and the corresponding Martingale difference $D_j := \mathcal{U}_j - \mathcal{U}_{j-1}$. Then we have

$$
\begin{aligned}
D_j &\leq \frac{1}{n+\hat{m}} \sup_{h \in \mathcal{H}} \left| w_t(y_i^{\max}) l(h(x_i^{\max}), y_i^{\max}) - w_t(y_i^{\min}) l(h(x_i^{\min}), y_i^{\min}) \right| \\
&\leq \frac{\alpha^* d_\infty(q\|t)}{n+\hat{m}}.
\end{aligned}
\qquad (47)
$$

In the following, we bound the conditional second moment $\mathbb{E}\left[D_j^2 | \mathcal{F}_{j-1}\right]$:

$$
\begin{aligned}
&\mathbb{E}\left[D_j^2 | \mathcal{F}_{j-1}\right] \\
&\leq \mathbb{E}\left[\left(\sup_{h \in \mathcal{H}} \frac{1}{n+\hat{m}} w_t(y_j^T) \hat{\pi}(x_j^T) l(h(x_j^T), y_j^T)\right)^2 \Big| \{(x_i^T, y_i^T)\}_1^{j-1}\right] P(\mathcal{C}_j | \mathcal{F}_{j-1}) \\
&\qquad + \left(\mathbb{E}\left[\frac{1}{n+\hat{m}} w_t(y_j^T) \Big| \{(x_i^T, y_i^T)\}_1^{j-1}\right]\right)^2 P(\mathcal{C}_j' | \mathcal{F}_{j-1}) \\
&\leq \frac{\alpha^* d(q\|t)}{(n+\hat{m})^2} P(\mathcal{C}_j | \mathcal{F}_{j-1}) + \frac{1}{(n+\hat{m})^2} P(\mathcal{C}_j' | \mathcal{F}_{j-1}) \\
&\leq \frac{\alpha^* d(q\|t)}{(n+\hat{m})^2}.
\end{aligned}
\qquad (48)
$$

The above inequality holds since $\alpha^* \geq 1$ and $d(q\|t) \geq 1$. Then using the Freedman's inequality, we have

$$P\left[\sum_{j=1}^{n+\hat{m}} D_j = \mathbb{E}\left[G_{n+\hat{m}}\right] - G_{n+\hat{m}} \geq \varepsilon\right] \leq \exp\left(\frac{-2\varepsilon^2}{2\left(\frac{\alpha^* d_\infty(q\|t)}{n+\hat{m}} + \varepsilon \frac{\alpha^* d(q\|t)}{n+\hat{m}}\right)}\right). \qquad (49)$$

Then we obtain the following generalization bound with probability at least $1 - \delta$

$$
\begin{aligned}
&\sup_{h \in \mathcal{H}} \left(\mathcal{L}(h; w_t) - \hat{\mathcal{L}}(h; w_t)\right) \\
&\leq 2\mathcal{R}_{n+\hat{m}}(\mathcal{G}(\mathcal{L}, \mathcal{H})) + \frac{2\alpha^* d_\infty(q\|t) \log(2/\delta)}{n+\hat{m}} + \sqrt{\frac{2\alpha^* d(q\|t) \log(2/\delta)}{n+\hat{m}}} \\
&= \varepsilon_{\mathcal{G}}(n+\hat{m}, d_\infty(q\|t), d(q\|t)).
\end{aligned}
\qquad (50)
$$

[3] Finally we analyze the upper bound of $\hat{\mathcal{L}}(h; w_t) - \hat{\mathcal{L}}(h; \hat{w}_t)$. We define that $\hat{l}(j) = \sum_{i=1}^{n+\hat{m}} \mathbb{I}_{y_i^T = j} \hat{\pi}(x_i^T) l(h(x_i^T), y_i^T)$ and $\|\hat{l}\|_2 \leq (n+\hat{m})\alpha^*$. Thus, we have the following conclusion by Cauchy Schwarz inequality:

$$
\begin{aligned}
\hat{\mathcal{L}}(h; w_t) - \hat{\mathcal{L}}(h; \hat{w}_t) &= \frac{1}{n+\hat{m}} \sum_{i=1}^{n+\hat{m}} \left(w_t(y_i^T) - \hat{w}_t(y_i^T)\right) \hat{\pi}(x_i^T) l(h(x_i^T), y_i^T) \\
&\leq \frac{1}{n+\hat{m}} \sum_{j=1}^{K} \left(w_t(j) - \hat{w}_t(j)\right) \hat{l}(j) \\
&\leq \frac{1}{n+\hat{m}} \|w_t - \hat{w}_t\|_2 \|\hat{l}\|_2 \\
&\leq \alpha^* \left((1 - \lambda)\|\theta\|_2 + \lambda\|\hat{\theta} - \theta\|_2\right).
\end{aligned}
\qquad (51)
$$

Similar to RLLS, we have that

$$||\hat{\theta} - \theta||_2 \leq \mathcal{O}\left(\frac{1}{\sigma_{\min}}\left(||\theta||_2\sqrt{\frac{\log(K/\delta)}{n+\hat{m}}} + \sqrt{\frac{\log(1/\delta)}{n+\hat{m}}} + \sqrt{\frac{\log(1/\delta)}{m}}\right)\right). \tag{52}$$

Thus we obtain the following generalization bound with probability at least $1 - \delta$

$$\begin{aligned}
&\hat{\mathcal{L}}(h; w_t) - \hat{\mathcal{L}}(h; \hat{w}_t) \\
&\leq \alpha^*\left(\lambda\mathcal{O}\left(\frac{1}{\sigma_{\min}}\left(||w-1||_2\sqrt{\frac{\log(K/\delta)}{n+\hat{m}}} + \sqrt{\frac{\log(1/\delta)}{n+\hat{m}}} + \sqrt{\frac{\log(1/\delta)}{m}}\right)\right) + (1-\lambda)||w-1||_2\right) \\
&= \alpha^*\varepsilon_w(n+\hat{m}, ||w-1||_2).
\end{aligned} \tag{53}$$

Combined with the above parts, the final conclusion final generalization bound is established with probability at least $1 - 3\delta$.

$$\begin{aligned}
\mathcal{L}(\hat{h}_{\hat{w}_t}; \tau) - \mathcal{L}(h^*; \tau) &\leq \varepsilon_{\mathcal{G}}(n+\hat{m}, d_\infty(q||t), d(q||t)) + \varepsilon_w(n+\hat{m}, ||w-1||_2) \\
&+ \frac{2\alpha^*rm}{n+rm}\left(d_{X|Y}(q||\hat{q}) + d_Y(q||\hat{q})\right) + 2\alpha^*||w_t(Y)||_\infty \text{err}(h_{\tilde{v}}).
\end{aligned} \tag{54}$$

## A.2 ADDITIONAL DESCRIPTION OF METHODS

In this section we introduce the pseudo-program of the ALS framework.

---

**Algorithm 1** Algorithm of ALS

---

**Input**: The labeled source set $\{x_i, y_i\}_{i=1}^n$ and the unlabeled target set $\{x_i\}_{i=n+1}^{n+m}$, pseudo-label selection ratio $r$, regularized parameters $\beta$.

**Initialize**: Initialize the source classifier $\hat{v} = \arg\min_v -\frac{1}{n}\sum_{i=1}^n\sum_{k=1}^K (y_i)_k \log \hat{p}(k|x_i, v)$, where $v$ is the parameter of $h$.

**Do**

    Use BBSE or RLLS to calculate the importance weight $\hat{w}$;

    Update the target classifier $\tilde{v} = \arg\min_v -\frac{1}{n}\sum_{i=1}^n\sum_{k=1}^K \hat{w}(k)(y_i)_k \log \hat{p}(k|x_i, v)$;

    Get the target pesudo labels with confidence

$$\hat{Y}^Q = \arg\min_{y_i} -\frac{1}{m}\sum_{i=n}^{n+m}\sum_{k=1}^K y_i^k \log \frac{\hat{p}(k|x_i, \tilde{v})}{\lambda_k}, \quad s.t.\ y_i^k \in \{0,1\},\ \forall i \in \{1, 2, ..., n\}.$$

    To align the conditional distribution, calculate the sample weights $\hat{\pi}$ by the following loss:

$$\min_\pi \sum_{k=1}^K \left(\frac{(\pi^k)^T\mathbf{K}^{\hat{Q}}\pi^k}{\hat{m}_k^2} - 2\frac{1^T\mathbf{K}^{P,\hat{Q}}\pi^k}{\hat{m}_k n_k} + \beta_k||\pi^k||^2\right), \quad s.t. \sum_{i=1}^{\hat{m}_k}\pi_i^k = \hat{m}_k,\ \pi_i^k \geq 0.$$

    Obtain the new training set $[\{x_i, y_i\}_{i=1}^n, \hat{\pi}(x_i)\{x_i, \hat{y}_i\}_{i=n}^{n+\hat{m}}]$;

    Update the new source classifier $\hat{v} = \arg\min_v -\frac{1}{n+\hat{m}}\sum_{i=1}^{n+\hat{m}}\hat{\pi}(x_i)\sum_{k=1}^K (y_i)_k \log \hat{p}(k|x_i, v)$;

    Use BBSE or RLLS to calculate the new importance weight $\hat{w}_t$;

    Update the target classifier $\tilde{v} = \arg\min_v -\frac{1}{n+\hat{m}}\sum_{i=1}^{n+\hat{m}}\hat{\pi}(x_i)\sum_{k=1}^K \hat{w}_t(k)(y_i)_k \log \hat{p}(k|x_i, v)$.

**End procedure**

**Output**: the new target classifier $\tilde{v}$.

---

### A.3 ADDITIONAL EXPERIMENTS

#### A.3.1 PREDICTIVE RESULTS

To show the effectiveness of ALS framework, we report the classification Accuracy(Acc) under the different proportions of pseudo label samples on the Tweak-one shift datasets. Due to the large number of CIFAR100 dataset categories, the number of most categories approaches 0 under Tweak-one shift. Thus, we do not think about Tweak-one shift CIFAR100 datasets. See the main text for the detailed result analysis.

Table 2: Acc comparison with different ratio $r$ on Tweak-one shift MNIST and CIFAR10 datasets.

| Methods | $\rho = 0.2$ | | | $\rho = 0.5$ | | | $\rho = 0.8$ | | |
|---|---|---|---|---|---|---|---|---|---|
| | $r = 0.1$ | $r = 0.3$ | $r = 0.5$ | $r = 0.1$ | $r = 0.3$ | $r = 0.5$ | $r = 0.1$ | $r = 0.3$ | $r = 0.5$ |
| MNIST | | | | | | | | | |
| WW | 0.8199 | 0.8199 | 0.8199 | 0.6386 | 0.6386 | 0.6386 | 0.5230 | 0.5230 | 0.5230 |
| CBST | 0.8562 | 0.8574 | 0.8615 | 0.7973 | 0.8012 | 0.7883 | 0.6604 | 0.6723 | 0.6645 |
| BBSE | 0.8493 | 0.8493 | 0.8493 | 0.7827 | 0.7827 | 0.7827 | 0.6722 | 0.6722 | 0.6722 |
| RLLS | 0.8501 | 0.8501 | 0.8501 | 0.7806 | 0.7806 | 0.7806 | 0.6902 | 0.6902 | 0.6902 |
| BBSE-ALS | 0.8630 | 0.8663 | 0.8714 | 0.8157 | 0.8197 | 0.8081 | 0.7673 | 0.7920 | 0.7780 |
| RLLS-ALS | 0.8660 | 0.8666 | 0.8718 | 0.8220 | 0.8204 | 0.8145 | 0.7892 | 0.8068 | 0.7873 |
| AvgIMP | **0.0148** | **0.0168** | **0.0219** | **0.0372** | **0.0384** | **0.0297** | **0.0971** | **0.1182** | **0.1015** |
| CIFAR10 | | | | | | | | | |
| WW | 0.3530 | 0.3530 | 0.3530 | 0.2689 | 0.2689 | 0.2689 | 0.2181 | 0.2181 | 0.2181 |
| CBST | 0.4266 | 0.4401 | 0.4435 | 0.3887 | 0.3912 | 0.3903 | 0.3327 | 0.3276 | 0.3058 |
| BBSE | 0.4092 | 0.4092 | 0.4092 | 0.3726 | 0.3726 | 0.3726 | 0.3204 | 0.3204 | 0.3204 |
| RLLS | 0.4124 | 0.4124 | 0.4124 | 0.3814 | 0.3814 | 0.3814 | 0.3366 | 0.3366 | 0.3366 |
| BBSE-ALS | 0.4356 | 0.4484 | 0.4544 | 0.4138 | 0.4195 | 0.4078 | 0.3880 | 0.3778 | 0.3634 |
| RLLS-ALS | 0.4381 | 0.4532 | 0.4625 | 0.4196 | 0.4233 | 0.4141 | 0.3911 | 0.3851 | 0.3788 |
| AvgIMP | **0.0261** | **0.0400** | **0.0477** | **0.0397** | **0.0444** | **0.0340** | **0.0611** | **0.0530** | **0.0426** |

#### A.3.2 INFLUENCE OF BASE CLASSIFIERS

In the main experiment, we use a two-layer fully connected neural network(MLP) for CIFAR10 datasets. In addition, in order to verify the impact of base classifier performance on model effect, ResNet-18(RES) is also used as the base classifier on CIFAR10 dataset.

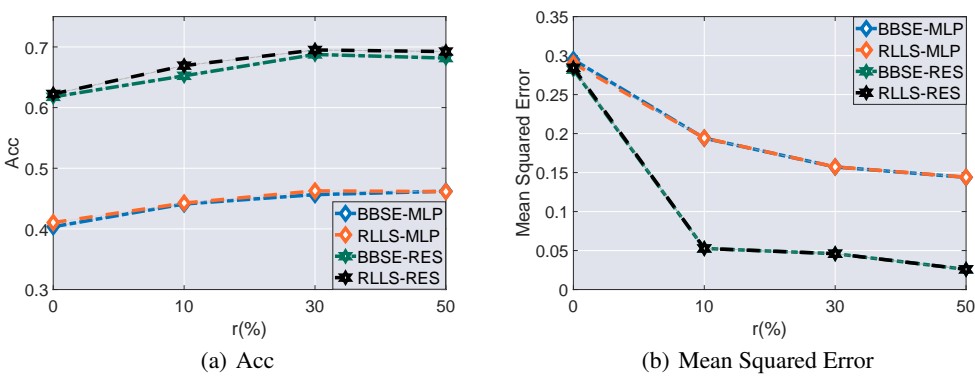

(a) Acc       (b) Mean Squared Error

Figure 3: The influence of base classifiers with on CIFAR10 dataset with $\alpha = 5$.

From the experimental results shown in Figure 3, we have the following observations: (1) No matter the quality of the base classifier, the ALS framework can improve the Acc and reduce the Mean

Squared Error compared with BBSE and RLLS. (2) Good base classifier can obtain better weight estimation than the common base classifier. This may be that a good classifier can more accurately estimate the confusion matrix and target label distribution.

### A.3.3 PARAMETER SENSITIVITY

Here, we check the sensitivity of the trade-off parameter $\beta \in \{0, 0.001, 0.1, 10, 1000\}$ of RLLS-ALS on MNIST and CIFAR10 datasets with $\alpha = 2$ and $r = 30\%$. The experimental results are shown in Figure 4. We find that its performance shows a trend of first increase and then decrease. If $\beta = 0$, the selected samples are rather sparse and thus the effect is mediocre. If $\beta = 1000$, the weight of all samples is almost always 1 which may not align the condition distribution. In our experiments, we find $\beta = 0.1$ works well on most datasets.

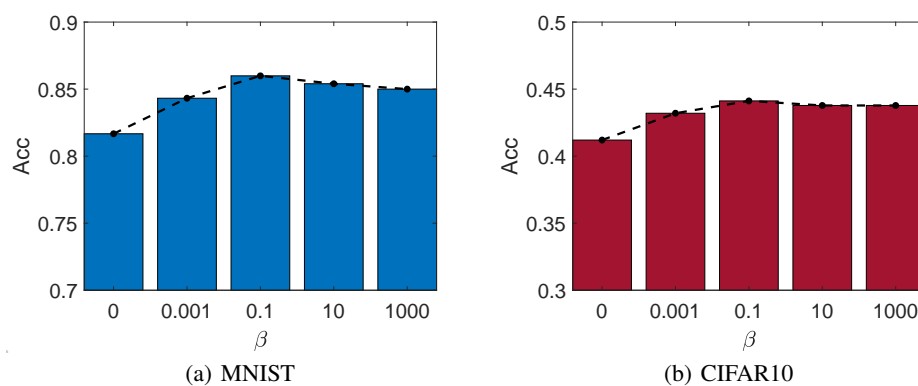

(a) MNIST        (b) CIFAR10

Figure 4: The parameter sensitivity of RLLS-ALS on MNIST and CIFAR10 datasets with $\alpha = 2$ and $r = 30\%$.

### A.3.4 COMBINATION WITH MLLS

ALS can be viewed as a additional trick for improving performance on most label shift methods, but there may be no theoretical guarantees. To verify the effectiveness of ALS, we use MLLS as the base method to estimate the importance weights and the proposed approach is called MLLS-ALS. We conduct experiments under different shift on MNIST and CIFAR100 datasets with $r = 30\%$, and the experimental results are shown in Figure 5. Through observation, the proposed method achieves good performance in most cases.

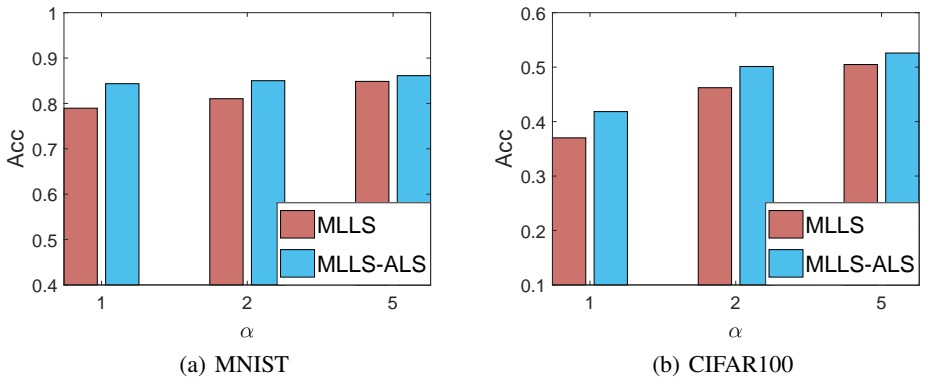

(a) MNIST        (b) CIFAR100

Figure 5: The performance comparison of MLLS and MLLS-ALS on MNIST and CIFAR100 datasets with $r = 30\%$.

A.3.5 TIME COST

To show the time cost of ALS, we compare the run time(in seconds) by BBSE-ALS and BBSE in Table 3. The run time is measured on a desktop computer with Intel(R) Xeon(R) Platinum 8163 CPU @ 2.50GHz × 2. The experimental results show that ALS does not have a particularly high time cost. If we pick more pseudo-labels, we will spend more time training the reweighted target classifier. However, we only select samples with the highest confidence that can maintain the label ratio in this paper, so it does not result in too much time cost.

Table 3: Average runtime comparison (seconds) on three datasets.

| Methods | MNIST | CIFAR10 | CIFAR100 |
|---|---|---|---|
| BBSE | 54.54 | 1630.91 | 12514.72 |
| BBSE-ALS | 169.17 | 3877.56 | 21951.07 |

