# OpenReview forum: "Alleviating Label Shift Through Self-trained Intermediate Distribution: Theory and Algorithms"
_ICLR.cc/2024/Conference — Submitted to ICLR 2024_

### Official Review · Reviewer_goif · 2023-10-16

**Soundness:** 3 good
**Presentation:** 4 excellent
**Contribution:** 3 good
**Rating:** 6
**Confidence:** 3

**Summary:**

This paper proposes an algorithm called ALS which uses intermediate distribution approximating the ideal distribution to alleviate label shift. It also gives the generalization guarantee of ALS under label distribution differences and pseudo-target labels.

**Strengths:**

1. The theory is sufficiently comprehensive, providing a solid guarantee for the rationality of algorithm design.

2. The method proposed is reasonable and performs well on three benchmark datasets.

3. The paper is well-written with clear logic.

**Weaknesses:**

In fact, I would like to give a score of 7 (weak accept) which is closer to the level of this paper, but there is no such option. The reason the score did not achieve 8 is that there is still room for improvement in experiments and analysis.

1. The three datasets, MNIST, CIFAR10, and CIFAR100, used to construct the experimental scenarios, are somewhat simplistic. Moreover, certain experiments and metrics are only disclosed for one or two of these datasets.
2. The disclosure of experimental hyperparameters appears to be incomplete. Algorithm hyperparameters, network hyperparameters, hyperparameters for comparison methods, random seeds, and other relevant details should be fully presented. Experimental results should include the mean and variance of metrics from multiple trials.
3. The two types of distribution changes in the experimental setup are not intuitive, lacking visualization, making it difficult for readers to perceive the extent of label distribution shift in the experimental setup. It is also challenging to determine what the most extreme scenarios included in the experimental setup are like.
4. The applicability of the algorithm should be thoroughly discussed. Considerations such as when the algorithm might fail, its compatibility with other algorithms and models, and its performance in extreme conditions (such as extreme imbalance or unseen categories) should be discussed.

The above points are just requirements for an 8-score paper. Of course, even if these issues are not addressed, I acknowledge that this paper is above the acceptance threshold.

**Questions:**

What type of F-score do you use for multi-class classification?

---

> ### Author Response · Authors · 2023-11-18
>
> Thank you for your comments.
> For the first weakness, since our source dataset has the restriction of label distribution, we try to verify the performance of ALS on a larger dataset with a larger number of categories. However, due to time constraints, we did not obtain new results. But, we add MLLS, which is an advanced method for last two years, as a comparison method to illustrate ALS more effectively.
> For the second weakness, we directly adopt the code provided by the authors of the compared methods. Since the comparison method can be embedded in our method, the network architecture and hyperparameters are set according to the compared methods. We illustrate it in the paper for a clear illustration.
> For the third weakness, we use the parameter $\alpha$ to determine the label distribution. We choose $\alpha=1$ as the upper bound of the shift, because $\alpha=0.5$ may cause some classes to have zero data, which will break the algorithm.
> For the forth weakness, the new class can be seen as a special case of extreme imbalance. One of the directions for future research is how to correct the model in the case of extreme imbalance. This article focuses more on how to use the data in the target domain, and thus does not consider extreme imbalanced cases specially.
> For the first question, we adopt micro-Fscore for multi-class classification.

---

> > ### Comment · Reviewer_goif · 2023-11-22
> >
> > Thank you for your response. It addressed my main concerns, and I will maintain my score.

---

### Official Review · Reviewer_MY42 · 2023-10-30

**Soundness:** 3 good
**Presentation:** 3 good
**Contribution:** 3 good
**Rating:** 6
**Confidence:** 4

**Summary:**

In order to efficiently deal with label shift, this paper propose to learn a self-trained intermediate distribution constructed from the labeled source and unlabeled target samples to approximate the ideal intermediate distribution, the sample complexity and generalization guarantees for the proposed approach is given. The introduction and theoretical analysis of the proposed method is clear.

**Strengths:**

The label shift is a widely existed problem and attract much attention. The intermediate distribution is used in this paper to address the label shift problem, which is interesting. From ideal intermediate distribution to self-trained distribution, the deeply analyze the issues of intermediate distribution, for example the pseudo-label, selection bias, and give the concrete solutions and the theorecital analysis. The organization is well and the presentation is clear.

**Weaknesses:**

The intermediate distribution is adopted in label shift problem, which may cause the expensive time cost.  The BBSE or RLLS is still used in the proposed method to estimate the pseudo-label, so the work can be considered as the combination of BBSE or RLLS and intermediate distribution.

**Questions:**

(1)	Since selection bias, the label conditional distribution may change, then the sample weight is adopted to address this issue. This is one of the key point in the proposed method. However, how to calculate the sample weight is not clear: the different notions are mixed-use, i.e., \pai\^ k and \pai\_k, and the same sample weight is assigned to samples in the same class? In Eqn. (10) the sample weight \pai\_i^k is used, however, in the context, only \pai\^k is discussed.
(2)	There exists some minor error, such as the “i” in Eqn.(7) is confusion with different range, and in Appendix, the No. of equation is wrongly used, for example, “Combining Eq.(17) and Eq. (20), we have”,” Combining Eq. (22), Eq. (24), Eq. (25) and Eq. (26), we have”. The method BBSE is wrongly written as BBLS in "which outperforms both BBSL and RLLS across diverse datasets and.......". In Table 1. the combination of BBSE or RLLS and ALS is wrongly written as BBSE-LSC or RLLS-LSC.
(3)	In Appendix, the “t(Y = y)” is used in Eqn.(7), this maybe a mistake.
(4) In the Introduction, MLLS is mentioned, which is one of the latest method for label shift. So it is better to show the experimental results about MLLS or combination of MLLS and ALS.

---

> ### Author Response · Authors · 2023-11-18
>
> Thank you for your comments.
> For the first weakness, we analyze the time cost through experiments in subsection A.3.5 of the supplementary material and the experimental results show that our method does not have a particularly high time cost. Although if we pick more pseudo-labels, we will spend more time training the reweighted target classifier. However, we only select samples with the highest confidence that can maintain the label ratio in this paper, so it does not result in too much time cost.
> For the first question, we revise this paper in the following two parts. (1) In view of the grammar and symbol usage problems of the article, we again checked and revised it to ensure that there were no symbol misuse and syntax error problems. (2) We propose a new method to extend ALS with MLLS, which is an advanced method for last two years, and call it MLLS-ALS. Then we compare MLLS-ALS with MLLS, and the experimental results in subsection A.3.4 of the supplementary material show that MLLS-ALS achieves good performance in most cases.

---

> > ### Comment · Reviewer_MY42 · 2023-11-23
> >
> > Thank you for your response, most of my issues have been addressed, the analysis about MLLS-ALS should be improved. I will maintain the rating.

---

### Official Review · Reviewer_zjuc · 2023-10-31

**Soundness:** 2 fair
**Presentation:** 2 fair
**Contribution:** 2 fair
**Rating:** 3
**Confidence:** 4

**Summary:**

The authors propose the ALS method for alleviating label shift. ALS works in conjunction with label shift methods BBSE or RLLS to find an "ideal intermediate distribution" between the source and target distributions to construct a modified train set used to retrain the classifier. Theoretically, the authors demonstrate the approach's sample complexity and generalization guarantees. Then, BBSE and RLLS with ALS are compared favorably in experiments to BBSE and RLLS without ALS, another approach CBST, and a model without any label shift technique applied.

**Strengths:**

- Method is easy to combine on top of BBSE or RLLS
- Proposed method performs well under the various examined data shift levels in the experiment, particularly in settings where data shift is very bad
- Approach is very theoretically grounded

**Weaknesses:**

- Proposed method is designed to be used in conjunction with methods that are several years old. Empirical evaluation also does not compare against more recent methods like the mentioned MLLS or other cited related works from last 2-3 years
- More space should be dedicated to theoretical results in the main body of the paper itself, especially since important choices (like using BBSE/RLLS) were made for the sake of getting good theoretical results
- Writing style could use a little revision to improve structure and clarity
   - Naming choices: By proposition 1, "ideal" intermediate distribution is non-unique, why not just call it an "intermediate distribution"? Acronym ALS is never explained (unless I just didn't see it)
   - Text should be checked by a proofreader or grammar software; eg "alleviates" in Definition 1 should take a grammatical object, unclear what's meant by "k-th class of label y" in beginning of 3.1 until reading the sentence after equation (8)

**Questions:**

- Because MLLS doesn't have good theoretical guarantees, I understand why you didn't want to do the theoretical analysis of ALS applied to MLLS and used BBSE/RLLS instead. However, can ALS still empirically be applied with MLLS or other more recent and well-performing methods? I could see selling ALS as a method-agnostic, additional trick for improving performance on top of most label shift techniques

---

> ### Author Response · Authors · 2023-11-18
>
> Thank you for your comments.
> For the first weakness, as the reviewers said, ALS can be viewed as an additional trick for improving performance on most label shift methods. Thus we propose a new method to extend ALS with MLLS, which is an advanced method for last two years, and call it MLLS-ALS. Then we compare MLLS-ALS with MLLS, and the experimental results in subsection A.3.4 of the supplementary material show that MLLS-ALS achieves good performance in most cases.
> For the second weakness, the reason why we use BBSE and RLLS as basis methods is to fancy their nice theoretical properties, which can be combined with ALS to obtain Proposition 7. If the basis method, such as MLLS, has no theoretical guarantees, it is difficult for us to obtain the desired theoretical properties. While there are no theoretical guarantees for MLLS, experiments can be conducted to illustrate the performance of ALS. The experimental results is shown in subsection A.3.4 of the supplementary material.
> For the third weakness, we illustrate the meaning of ALS in the part of Introduction, i.e., 'our approach, abbreviated as ALS, learns a self-trained intermediate distribution...'. We used 'ideal' earlier to better distinguish between the true and pseudo-label-guided intermediate distributions. Borrowing from the reviewer's suggestion, we drop the 'ideal' for clearer presentation. In view of the grammar problems of the article, we have checked and revised it again.
> For the first question, we compare the proposed MLLS-ALS with MLLS in subsection A.3.4 of the supplementary material to show the effectiveness of ALS.

---

> > ### Comment · Reviewer_zjuc · 2023-11-19
> >
> > Thank you for your response. I appreciate your work to improve the writing and especially in adding the experiment with MLLS-ALS.
> >
> > I think it's extremely reasonable to do the theoretical analysis using BBSE/RLLS since they already have good theory. However, I don't think this is a good reason to focus your empirical analysis on ALS using these methods. Instead, I think the paper should present empirical analysis of BBSE/RLLS/MLLS equally so you can emphasize ALS works with a lot of different methods.
> >
> > I now see ALS stands for Alleviating Label Shift; you could maybe try bolding/italicizing this phrase so people don't miss it, but this is a minor detail. The latest paper draft's edits have begun to improve the writing and math notation clarity, but I think it still needs more of this.
> >
> > For these reasons, I think ALS has promise but the paper could just use a little more time to develop. For these reasons I will maintain my rating of 3.

---

### Official Review · Reviewer_dogK · 2023-11-01

**Soundness:** 1 poor
**Presentation:** 1 poor
**Contribution:** 1 poor
**Rating:** 3
**Confidence:** 4

**Summary:**

The paper introduces an ideal intermediate distribution instead of the source distribution to reduce the variation in the target label distribution. The authors propose an algorithm to learn a self-trained intermediate distribution constructed from the labeled source and unlabeled target samples. Besides, the authors show the sample complexity and generalization guarantees for the proposed approach. The paper also includes extensive experimental results to support the main results.

**Strengths:**

1. Label shift is an interesting and valuable topic in the learning community.

2. The literature part is clear.

3. There is extensive experiment analysis on the algorithm accuracy performance.

**Weaknesses:**

Major
1. The paper needs to be more reader-friendly. A lot of notations are used without/before definition. This situation makes it very hard to understand the idea and review the solidness of the main results. For example, in only half of a page, more than10 notations are not clear:

1.1 P3, 3rd paragraph, 3rd line : \hat{q}_h is not shown how to get it.

1.2 P3, 3rd paragraph, 5th line: \gamma is not defined

1.3 P3, 3rd paragraph, 5th line: \theta is not defined

1.4 P3, 3rd paragraph, 5th line: \Delta_C is not defined

1.5 P3, Collaray 1, 2nd line: delta has no domain

1.6 P3,  Collaray 1, (3): q||p is not defined

1.7 P3, Collaray 1, (3) : \gamma is not defined and has no domain

1.8 P3, Collaray 1, (4): q||t is not defined

1.9 P3, Collaray 1, (4):  \R is not defined

1.10 P3, Collaray 1, (4):  \G is not defined

1.11 P3, Collaray 1, (4):  \H is not defined

1.12 P3, Collaray 1, (5):  \O is not defined

1.13 P3, Collaray 1, (5):  \sigma_{min} is not defined


2.The experimental section doesn't include the time performance.

**Questions:**

1. What's the meaning of those undefined notations?

---

> ### Author Response · Authors · 2023-11-18
>
> Thank you for your comments.
> For the first weakness, since these notations are explained by the cited literature and also used in Proposition 7, for reasons of space, we only explain these notations in the proof of the supplementary material. Taking into account the suggestions of the reviewers, we give instructions on the first use of these notations for a clearer illustration.
> For the second weakness, we analyze the time cost through experiments in subsection A.3.5 of the supplementary material and the experimental results show that our method does not have a particularly high time cost. Although if we pick more pseudo-labels, we will spend more time training the reweighted target classifier. However, we only select samples with the highest confidence that can maintain the label ratio in this paper, so it does not result in too much time cost.
> For the first question, we reinterpret these notations in the main text.

---

> > ### Comment · Reviewer_dogK · 2023-11-23
> >
> > Thanks for your reply.
> >
> > I have checked the updated manuscript for the notations, and many of them are still not defined when they first appear. The added table does not show the baseline running time for the time issue.
> >
> > Most of my concerns are not addressed, so I keep my score to 3.

---

> > > ### Author Response · Authors · 2023-11-23
> > >
> > > Thank you for your comments. The above notations have mentioned in Eq. (1)  and Corollary 1. We hope the reviewers can clarify which symbols have not been mentioned, and we will try our best to revise them. For the timing problem, we take the time of BBSE as the baseline and then show the time of our method BBSE-ALS, which can show the time taken by ALS.

---

### Meta-Review · Area_Chair_dHfN · 2023-12-06

**Metareview:**

This paper proposes to utilize a self-trained intermediate distribution to alleviate the label shift. The main idea is to bring the source data distribution closer to the target by using pseudo-label training hence the resulting new important weights are easier to estimate. Theoretically, the authors demonstrate the approach's sample complexity and generalization guarantees. Empirically, the method can be applied to most existing label shift correction methods to further improve performance.

Strength: all reviewers agree that the idea is interesting and the technique is sound.

Weakness: the main concern after rebuttal is the comparison/relation with previous work and the presentation.

No reviewer mentioned this but the intermediate distribution idea is very similar to this paper: Active learning under label shift. In The 24th International Conference on Artificial Intelligence and Statistics(AISTATIS), pp. 3412– 3420, 2021. Instead of looking at active learning, self-training, where data selection is essential, is considered here. This should be discussed further. The interaction with more recent label shift correction work, like MLLS, should also be incorporated into the main paper.

**Justification For Why Not Higher Score:**

Given the current presentation, the paper can benefit from another round of updates.

**Justification For Why Not Lower Score:**

N/A

---

### Decision · Program_Chairs · 2024-01-16

Reject